# Prompt-Driven Exploration

Sunshine Jiang*, John Marangola*, David Zhang*, Raghuram Kowdeed†,
Ruiyang Luo*, Nitish Dashora*, Richard Li*, Pulkit Agrawal*, Zhang-Wei Hong‡

*Massachusetts Institute of Technology
†University of California, Berkeley
‡MIT-IBM Watson AI Lab

*Abstract*—Exploration is essential to RL since a policy cannot improve by repeatedly sampling the behaviors it already prefers. Standard methods inject stochasticity in the action space, but such jitter only yields rollouts close to the original. Escaping a weak policy often requires global perturbations that action noise cannot produce. Large language models (LLMs) and vision-language-action (VLA) models offer a pathway: they condition the policy on a natural language prompt, and since the rollout follows from it, modifying the prompt induces global changes. The challenge is finding prompts that induce useful global changes. With a weak policy that rarely succeeds, reward is too sparse to select on. Our idea is to refine prompts from the rollouts themselves: a vision-language model (VLM) reasons over the rollout video, diagnoses how the policy responded, and rewrites the prompt to elicit better behavior next time. This procedure realizes posterior sampling, a classical RL exploration framework, at the level of prompts: the VLM maintains an implicit distribution over useful prompts and updates it from observed rollouts. We call this strategy Prompt-Driven Exploration (PDE). Across manipulation and reasoning tasks, PDE enables RL to learn successful policies even from zero-reward starts, and improves sample efficiency more broadly.

## I. Introduction

Reinforcement learning (RL) [20] has become a dominant post-training paradigm for foundation models because it enables scalable self-improvement beyond supervised learning. RL fine-tuning has unlocked strong reasoning and mathematical capabilities in large language models (LLMs) [18, 14] and enabled direct alignment with human preferences in large diffusion-based text-to-image models [6]. However, self-improvement is bottlenecked by exploration: a policy can only surpass its current behavior by producing rollouts different from those it already favors, so that RL can reinforce the higher-reward ones.

Standard practice perturbs the policy in action space by sampling actions stochastically rather than greedily [46, 36, 15]. But such action-level noise only jitters individual actions and yields rollouts close to the original; it induces only local exploration [33, 13], and the set of rollouts reachable by step-wise noise shrinks rapidly with the horizon and action dimension. Escaping a weak policy often requires global perturbations that alter behavior across the entire rollout, which action noise cannot produce. This limitation is especially pronounced in settings without a strong warm start, particularly vision-language-action (VLA) model fine-tuning on manipulation, where state-of-the-art models often start at near-zero success rates [22, 5].

If action noise only produces local exploration, how can we perturb the policy globally? Foundation models offer a pathway. LLMs [7, 1] and VLAs [22, 5] are conditioned on a natural language prompt, and since the entire rollout follows from it, modifying the prompt induces global changes. Figure 1 illustrates this on a microwave-closing task: under the prompt "close the microwave," the policy reaches for the mug and fails, and action noise only jitters the arm without changing strategy; rephrasing the prompt to "push on the microwave door until it shuts" redirects the policy to the correct contact point and succeeds, without any weight updates. Prior work on context engineering and prompt optimization has shown that prompt phrasing can significantly shape model behavior [7, 31, 21, 16], but prompts have not, to our knowledge, been used as an axis for exploration in RL.

The open question is how to find prompts that induce useful global changes. A natural answer is to try candidates and keep those that yield higher reward, but with a weak policy that rarely succeeds, reward is too sparse to select on. We instead refine prompts from the rollouts themselves: a vision-language model (VLM) reasons over the rollout video, diagnoses how the policy responded, and rewrites the prompt to elicit better behavior next time. This procedure mirrors posterior-sampling RL at the prompt level [39, 32, 10, 33]. A VLA defines a family of prompt-conditioned policies, so a distribution over prompts implicitly induces a distribution over policies, structured by the pretrained language prior. The VLM acts as an amortized posterior update: it samples plausible prompts from its language prior [52, 56, 38] and refines this distribution by reasoning over observed trajectories [47, 17, 42], without gradient training. This aligns with evidence that in-context learning can behave as implicit Bayesian inference [49, 44, 3]. We call the resulting algorithm Prompt-Driven Exploration (PDE): a VLM iteratively updates a prompt distribution from observed trajectories, and each rollout is generated by sampling a prompt from this distribution.

Our contribution is a simple exploration strategy that enables RL to escape weak initial policies by refining prompts from policy rollouts. We evaluate PDE on LIBERO [26] and LIBERO-PRO [55], using VLAs trained on only a fraction of the demonstrations. In this regime, the policy has no successful rollouts to reinforce, and action-space perturbations rarely discover success by chance. Across tasks of varying difficulty,

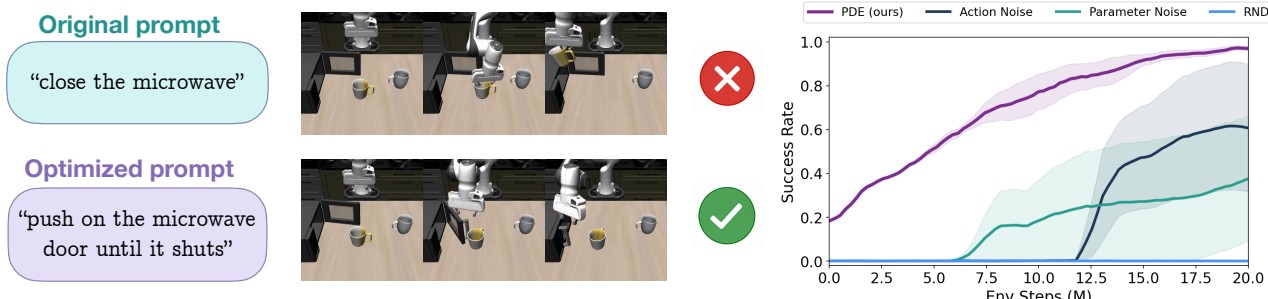

Fig. 1: **Left:** The same VLA policy fails under "close the microwave" by grasping the mug instead of pushing the door (top row), but succeeds when the prompt is rephrased to "push on the microwave door until it shuts" without any weight updates (bottom row). **Right:** RL with our prompt-driven exploration (PDE) bootstraps from $0\%$ to $98\%$ success, while standard action-noise exploration remains near zero.

PDE achieves higher success rates with far fewer environment interactions and solves tasks where action-space exploration fails. Our analysis shows that VLM updates reliably shift the prompt distribution toward prompts that elicit success, supporting the posterior-sampling view behind PDE. To show the generality of PDE beyond VLA control, we further evaluate it on challenging LLM coding tasks, where it also improves sample efficiency.

## II. RELATED WORKS

**RL post-training for VLAs.** Fine-tuning pretrained VLAs under sparse binary rewards is challenging at scale [40, 27, 24]. Recent work reduces what RL must update: freezing the backbone and training a small module on top, such as an RL token with actor-critic head [50] or residual off-policy actors distilled back into the base policy [48]; or optimizing in the latent space of a pretrained diffusion policy [43]. These change *how* the policy is updated; we change *what* it is asked to do.

**Foundation reward models.** A complementary line densifies the reward by training text-conditioned progress models, either as goal-reaching value functions [28, 29], from interpolated progress labels on demonstrations [30, 25, 53], or from preferences [45]. These reshape the reward for a given rollout distribution; PDE changes the rollout distribution itself, and the two are compatible.

**Prompt optimization.** LLMs can serve as prompt optimizers that iteratively refine prompts [52, 56], and analogous methods exist for VLMs through visually grounded discrete optimization [12] and continuous context embeddings [54]. Closest to our setting, CoVer-VLA [23] uses a trained verifier to optimize VLA instructions at test time. We instead use prompt diversity as an exploration mechanism during training.

**RL exploration strategies.** Classical alternatives to action noise include count-based bonuses [4], curiosity-driven intrinsic motivation [34], entropy regularization [15], and parameter-space noise [35]. These see limited adoption in VLA post-training, where auxiliary density or forward models over high-dimensional visual inputs are costly to train alongside billion-parameter policies. PDE explores in the space of task specifications instead, leveraging the language conditioning already present in pretrained VLAs.

## III. PRELIMINARIES

We fine-tune a vision-language-action (VLA) policy [5, 22] with reinforcement learning (RL) [20] in a multi-task setting. The standard RL pipeline trains a policy $\pi_\theta$ on a distribution of hand-crafted tasks. Each task $g$ is paired with a canonical prompt $p_g$. At the start of each rollout, a task $g$ is sampled and a prompt $p$ is given to $\pi_\theta$. At each timestep $t$, the policy receives the environmental observation $o_t$ (e.g., text, RGB images, etc) and outputs an action $a_t$ until horizon $T$. We denote the resulting rollout by $\tau_p = (o_0, a_0, \ldots, o_{T-1}, a_{T-1}, o_T)$. The policy receives a terminal reward

$$R(\tau_p, g) = \begin{cases} 1, & \text{if } \tau_p \text{ completes task } g, \\ 0, & \text{otherwise.} \end{cases}$$

**Reward depends on the task, not the prompt.** The reward $R$ is a function of the task $g$ and rollout $\tau_p$, but not of the prompt $p$. For example, if $g$ is "open the drawer," any rollout that opens the drawer receives reward 1, whether the prompt is the canonical prompt $p_g =$ "open the drawer" or an alternative such as "pull the handle toward you." Thus, a prompt need not faithfully describe $g$ to be useful. It only needs to condition $\pi_\theta$ toward a rollout that completes $g$. Section IV shows how our method exploits this flexibility for exploration. The RL objective is to maximize expected success:

$$J(\pi_\theta) = \mathbb{E}_{\substack{g \sim p_\mathcal{G},\, p \sim \rho(\cdot|g) \\ \tau_p \sim \pi_\theta(\cdot|p)}} \left[ R(\tau_p, g) \right]. \tag{1}$$

## IV. METHOD: PROMPT-DRIVEN EXPLORATION (PDE)

Standard RL explores by sampling actions stochastically from $\pi_\theta(\cdot \mid o_t, p_g)$ around the canonical prompt $p_g$, perturbing behavior locally at each timestep. PDE introduces a complementary axis of exploration: the prompt $p$ itself. Recall from Section III that the reward $R(\tau_p, g)$ depends only on the task $g$ and the resulting trajectory, not on the prompt that produced it. Any prompt that drives $\pi_\theta$ to complete $g$ therefore yields a successful rollout, regardless of whether its wording matches

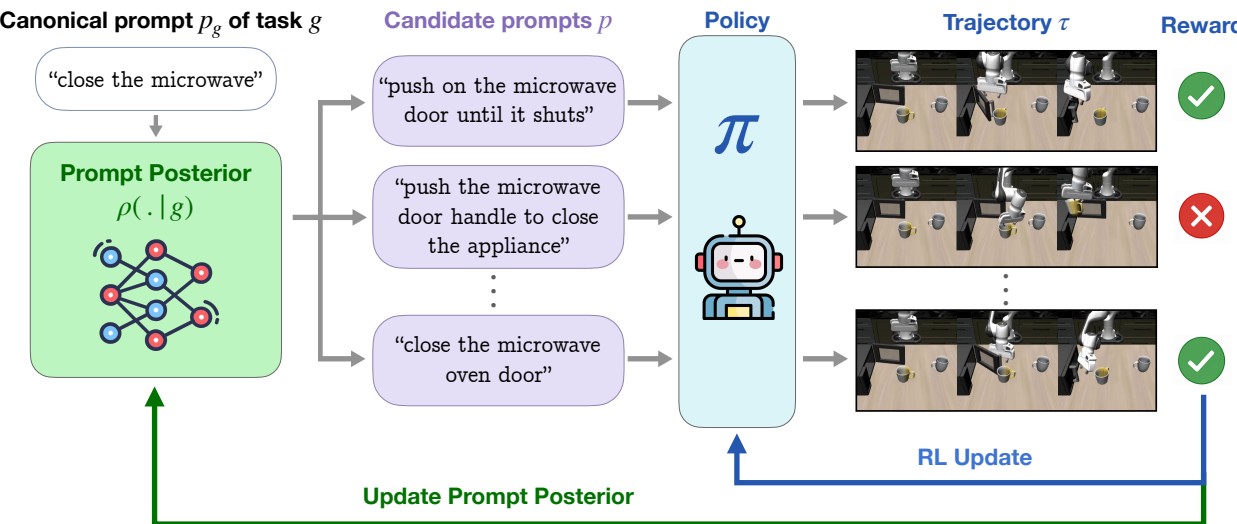

**Fig. 2:** Overview of our method. A VLM defines a prompt sampler $\rho(\cdot \mid g, \mathcal{H})$ over candidate prompts for task $g$, given its canonical prompt $p_g$ and previous rollout feedback. Prompts sampled from $\rho$ are executed by the VLA policy $\pi_\theta$. The resulting trajectories and rewards are added to the history, allowing $\rho$ to propose prompts that better elicit rollouts completing the task $g$ (green arrow, "Update Prompt Posterior"). Finally, $\pi_\theta$ is fine-tuned with RL on the collected rollouts (blue arrow, "RL Update").

$p_g$. Because $\pi_\theta$ conditions on $p$ at every timestep, replacing $p_g$ with an alternative shifts the policy's action distribution globally, exploring trajectories that step-wise action noise may not reach.

**Overview.** The challenge in exploration in the prompt space is selecting informative prompts: ones whose rollouts reveal how the policy responds and suggest how to refine the prompt next time. PDE addresses this in two stages. First, we search for useful prompts by rolling out the initial policy under candidates drawn from a broad distribution $\rho$ and using a VLM to analyze the resulting trajectories—diagnosing how the policy responded to each prompt and rewriting candidates accordingly—to update $\rho$ toward prompts that better elicit the intended behavior (**Update Prompt Posterior**, Fig. 2). Once $\rho$ has concentrated on such prompts, we fine-tune $\pi_\theta$ on rollouts sampled under $p \sim \rho$ via RL (**RL Update**, Fig. 2). Algorithm 1 summarizes the procedure.

### A. Posterior Sampling over Prompts

**From PSRL to prompt-space exploration.** Posterior sampling for reinforcement learning (PSRL) [39, 32] maintains a posterior over policies. At iteration $i$, PSRL samples one policy, executes it for one episode—from the initial state to termination or horizon $T$—and updates the posterior using the observed trajectory and reward. This produces consistent exploration: all actions in the episode are chosen by the same sampled policy, rather than being perturbed independently at each timestep. The challenge is scalability. For modern VLAs, the policy is a large neural network $\pi_\theta$, making a Bayesian posterior over parameters intractable to maintain and sample from.

**Prompts induce a tractable policy posterior.** PDE recovers the core idea of PSRL through the language interface of a pretrained VLA. For fixed VLA parameters $\theta$, each prompt $p$ induces a policy

$$\pi_p(\cdot \mid o) := \pi_\theta(\cdot \mid o, p).$$

Thus, a distribution over prompts induces a distribution over policies. In this induced policy class, drawing a policy hypothesis amounts to choosing a prompt. Because the prompt is fixed throughout the rollout, it can shift the behavior attempted in the episode while preserving temporal coherence. The uncertainty is no longer over independent low-level action perturbations, but over which prompt will elicit successful behavior from the VLA.

**A VLM as an implicit prompt posterior.** PDE represents this prompt distribution with a VLM prompt sampler $\rho$. Let

$$\mathcal{H}_i = \{(g_j, p_j, \tau_j, R(\tau_j, g_j))\}_{j < i}$$

denote the interaction history before iteration $i$. PDE samples $p_i \sim \rho(\cdot \mid g_i, \mathcal{H}_i)$, where $\rho$ conditions on the task, previous prompts, trajectories, and rewards. The prompt posterior is updated implicitly by appending each new rollout to $\mathcal{H}_i$, as shown in Algorithm 1.

Unlike a classical Bayesian posterior, $\rho(\cdot \mid g, \mathcal{H}_i)$ has no explicit density over natural language. It is instead queried as an implicit sampler: given the task and rollout feedback, it proposes the next prompt to try. This lets PDE use the VLM's language and vision priors to propose coherent, task-relevant prompts, while trajectory feedback adapts those proposals to the current VLA.

**Algorithm 1** Prompt-Driven Exploration (PDE).

---

**Require:** VLA policy $\pi_\theta$; task distribution $\mathcal{G}$; VLM prompt sampler $\rho$; mixture schedule $\{\alpha_i\}_{i=1}^{N}$; iterations $N$
 1: Initialize history $\mathcal{H}_1 \leftarrow \emptyset$
 2: **for** $i = 1, \ldots, N$ **do**
 3:     Sample a task $g_i \sim \mathcal{G}$ with canonical prompt $p_{g_i}$
 4:     Sample a prompt $p_i \sim \alpha_i \delta_{p_{g_i}} + (1 - \alpha_i)\rho(\cdot \mid g_i, \mathcal{H}_i)$
 5:     Roll out $\pi_\theta(\cdot \mid o, p_i)$; observe $\tau_i$ and $R(\tau_i, g_i)$
 6:     Update history $\mathcal{H}_{i+1} \leftarrow \mathcal{H}_i \cup \{(g_i, p_i, \tau_i, R(\tau_i, g_i))\}$
 7:     Update $\theta$ with PPO on $\tau_i$ using mixed backpropagation between $p_i$ and $p_{g_i}$
 8: **end for**
 9: **return** $\pi_\theta$

---

### B. Combining PDE with Policy Optimization

We implement the RL update in Algorithm 1 with Proximal Policy Optimization (PPO) [36]. For a rollout $\tau = \{(o_t, a_t)\}_{t=0}^{T-1}$ collected under prompt $p$, PPO minimizes

$$\mathcal{L}^{\mathrm{PPO}}(\theta) = -\mathbb{E}_t\Big[ \min\big(w_t(\theta)\hat{A}_t, \, \mathrm{clip}(w_t(\theta), 1-\epsilon, 1+\epsilon)\hat{A}_t\big)\Big],$$
$$w_t(\theta) = \frac{\pi_\theta(a_t|o_t,p)}{\pi_{\mathrm{old}}(a_t|o_t,p)}, \tag{2}$$

where $\hat{A}_t$ is the advantage estimate. The main issue is that PDE may collect data under exploratory prompts $p \sim \rho(\cdot \mid g, \mathcal{H})$, while evaluation uses the canonical prompt $p_g$. We use two modifications to make exploratory rollouts improve the canonical-prompt policy.

**Mixture sampling.** During data collection, we sample $p \sim \alpha \delta_{p_g} + (1 - \alpha)\rho(\cdot \mid g, \mathcal{H})$. Prompts from $\rho$ encourage exploration, while prompts from $p_g$ keep training aligned with evaluation. We increase $\alpha$ as success under $p_g$ improves, shifting training toward the canonical prompt $p_g$.

**Mixed backpropagation.** A rollout collected under an exploratory prompt $p$ directly trains $\pi_\theta(\cdot \mid o, p)$, but may not improve $\pi_\theta(\cdot \mid o, p_g)$. To couple the two prompts, we replace the current-policy log-probability in the PPO ratio with the average log-probability under both prompts:

$$\ell_t(\theta) = \frac{1}{2}\log \pi_\theta(a_t \mid o_t, p_g) + \frac{1}{2}\log \pi_\theta(a_t \mid o_t, p),$$
$$w_t(\theta) = \exp(\ell_t(\theta) - \log \pi_{\mathrm{old}}(a_t \mid o_t, p)). \tag{3}$$

We then use this modified $w_t(\theta)$ in the PPO loss. Thus, an action receives a strong update only when it is likely under both the exploratory prompt $p$ and the canonical prompt $p_g$. The mixture schedule, trajectory summarization protocol, and other hyperparameters are given in Appendix A.

### V. Experiments

We structure the experiments around three questions: (1) we test whether PDE can find prompts that make a weak policy achieve non-zero reward and whether these prompts induce global behavior changes rather than local action noise (Section V-B), (2) we evaluate whether PDE enables RL fine-tuning from weak VLA initializations with near-zero success

rates on LIBERO-PRO and ManiSkill (Sections V-C and V-D), with ablations in Section V-F, and (3) we demonstrate that PDE also applies beyond robotics by using it on challenging LLM coding tasks.

### A. Setup

**Environment and Metric.** We evaluate on LIBERO [26] and ManiSkill [41]. We report `success_once`, the binary episode-level success metric used by the RLinf evaluation suite[1], averaged over 250 evaluation environments per task. We start RL training with an initial policy from a weak Pi05 SFT checkpoint [9] trained with a limited number of demonstrations from four LIBERO task suites, excluding LIBERO-90, unless specified.

**Baselines.** We compare PDE against six baselines covering three standard ways to improve exploration or learning in PPO: adding noise, adding auxiliary exploration bonuses, and using dense reward-model feedback. All methods start from the same supervised fine-tuning (SFT) checkpoint, use the canonical task prompt $p_g$ for training, and are trained with the same PPO [36] optimizer settings and rollout budget. Checkpoint details and implementation hyperparameters are provided in the corresponding result sections and Appendix E0b.

- **Noise**: **Action Noise** is standard PPO with stochastic action sampling. **Parameter Noise** applies Gaussian perturbations [35] to the policy weights at the beginning of each rollout.
- **Bonus**: Random Network Distillation (RND) [8] adds intrinsic reward, computed as the prediction error of a learned predictor against a fixed random target network.
- **Dense reward**: **Robometer** [25] provides dense task-progress rewards from egocentric images and the task description.

### B. Illustrative Experiment: Close the Microwave

**Setup.** We first examine a single LIBERO-90 task with a visually clear failure mode: $g = $ *"close the microwave"* (Figure 1, left). The canonical prompt is $p_g = g$. We start with an initial policy from a weak Pi05 SFT checkpoint [9] trained with a limited number of demonstrations from four LIBERO task suites, excluding LIBERO-90. As shown in Figure 1 (right), PPO with standard action-space exploration remains near zero and reaches only $\sim 10\%$ success by step 100, indicating that local action perturbations are inefficient for escaping this initial policy.

**Why does the initial policy fail?** The checkpoint exhibits a consistent failure mode across rollouts (Figure 1, top): the arm grasps the yellow-handled mug, moves it toward the microwave, and stalls without contacting the microwave door. This behavior is consistent with a visually similar training task, *"put the yellow and white mug in the microwave and close it."* The shared scene and the word *"microwave"* appear to bias the policy toward a pick-and-place trajectory, although the target task only requires closing the door. Thus, the canonical prompt

---

[1]https://rlinf.readthedocs.io/

elicits the wrong motor program. Action noise perturbs this behavior only locally, so most sampled rollouts still manipulate the mug and fail.

**Can PDE discover successful prompts?** We next run the prompt-posterior update stage of PDE on the fixed initial policy, before any policy weight updates. Within 85 cumulative rollouts, PDE discovers 10–12 unique prompts with nonzero success across runs (Figure 6). The successful prompts fall into three interpretable categories:

- **Explicit contact and action**: prompts such as *"push the microwave door closed," "push the microwave door until it clicks,"* and *"swing the microwave door shut"* specify the desired physical interaction, shifting the policy from pick-and-place behavior to push-and-close behavior.
- **Spatial or generic reference**: prompts such as *"close the black appliance door on the left," "close the kitchen appliance door,"* and *"shut the open appliance door"* identify the target by appearance or location rather than by the object name *"microwave,"* reducing reliance on the misleading object-name cue.
- **Distractor exclusion**: prompts such as *"do not close the cabinet – close microwave"* and *"first move to the microwave, then close the door"* explicitly redirect the policy away from distractors and toward the intended contact point.

These prompts show that the VLM can use rollout feedback to infer how the VLA interprets previous prompts and propose alternatives that better elicit the desired behavior. However, prompt optimization alone is insufficient: the best discovered prompts remain below 40% success. Language can reveal a more useful behavior mode, but RL is needed to turn it into a successful policy.

**Can PDE bootstrap RL from zero success?** We then fine-tune the policy with PPO using PDE rollouts. Because PDE samples from the learned prompt distribution $\rho(\cdot \mid g, \mathcal{H})$, training receives nonzero-reward trajectories even though the canonical prompt $p_g$ initially has 0% success. As shown in Figure 1 (right), PDE starts at $\sim 20\%$ success, reflecting the successful exploratory prompts, and reaches near 100% success by step 100 under the canonical prompt $p_g$. In contrast, Action Noise barely escapes zero, RND fails to make progress, and Parameter Noise improves only partially with high variance.

**Summary.** This case study illustrates the core mechanism of PDE: prompt-space exploration discovers globally different rollouts that action-space perturbations rarely reach, and PPO transfers these successful behaviors back to the policy under the canonical prompts $p_g$.

### C. Benchmark Results on LIBERO-PRO

**Setup.** We evaluate PDE on LIBERO-PRO [55], a benchmark that modifies LIBERO environments and tasks to create challenging generalization settings where current VLA models often struggle. This makes LIBERO-PRO well suited for testing whether an exploration method can fine-tune a policy from a weak initialization. We evaluate on 120 tasks spanning four LIBERO task suites. The benchmark includes many tasks where the initial policy has near-zero success, while also covering easier tasks where action-space exploration can make progress. This range lets us compare exploration methods across different levels of initial policy competence. We exclude *language perturbation* because PDE also modifies prompts, which would confound the comparison, and *environment perturbation* because its BDDL files are not publicly released. We train separate policies for each benchmark group; details are provided in Appendix E0b.

**Does PDE improve over standard action noise?** Since our method is designed primarily to help when the base policy provides little or no reward signal, we group the 120 tasks into three difficulty tiers by the initial policy's success rate to test whether the gains concentrate on the hardest tasks as predicted: *hard* (init SR < 10%; 47 tasks), *medium* (10–80%; 36 tasks), and *easy* ($\geq 80\%$; 37 tasks). Figure 3 and Table VII report success rates aggregated by tier. Our method outperforms all the baselines across every difficulty tier, with the largest gap on the *hard* tier (60% relative improvement), where many tasks have zero initial success and action-noise methods receive no learning signal. This is consistent with the microwave case study (Section V-B): PDE discovers language groundings that unlock nonzero performance on tasks where the initial policy has overfit to a wrong motor program, precisely the regime in which action-space perturbations are ineffective.

**Does PDE generalize across base models?** We repeat the difficulty-aggregated benchmark on GR00T and Pi0 (Table VIII, Figures 4–5). PDE consistently improves over Action Noise across both backbones. On GR00T, PDE improves learning across all difficulty groups, with the largest gains on hard and medium tasks. On Pi0, PDE also improves hard and medium tasks, while the easy-task curves are closer because both methods already start from high success. For Pi0 hard tasks, the benefit is most visible in the learning curve: PDE maintains a consistent advantage throughout training, although the absolute success rates remain low, making aggregate table gains appear smaller. We use the same PDE procedure for both models, but choose the backpropagation variant separately.

### D. Benchmark Results on ManiSkill

**Setup.** Following [27], we evaluate generalization on ManiSkill `PutOnPlateInScene25` [41], a tabletop pick-and-place benchmark where the policy must place one of 25 objects onto a receptacle in a kitchen scene. The Pi0.5 SFT checkpoint is trained only on the in-distribution split, which contains 16 objects and 16,384 episodes synthesized with MPLib[2]. We fine-tune one policy per OOD variant with PPO, evaluating 12 variants across three generalization scenarios:

- **Vision** (5 variants): novel backgrounds and textures induced by image overlays of varying intensity.
- **Semantics** (4 variants): unseen objects, varied instructions, and distractors such as extra objects.
- **Execution** (3 variants): varied initial states, unseen robot poses, and dynamic disturbances.

[2]https://github.com/haosulab/mplib

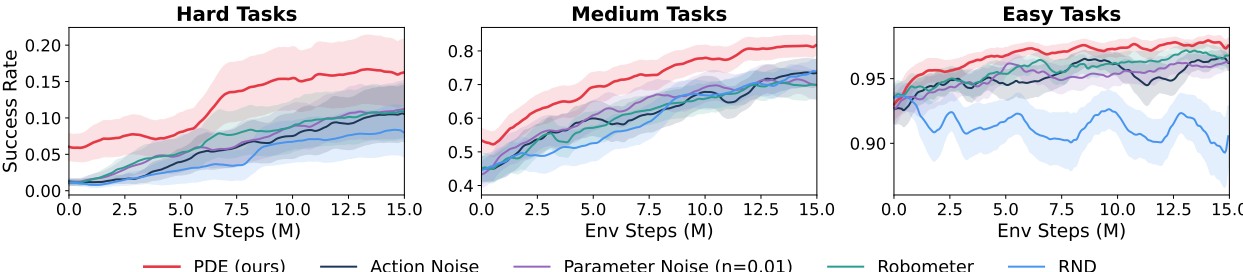

Fig. 3: Aggregated training curves grouped by task difficulty based on initial success rate. **Left:** Hard tasks (initial Success Rate $< 0.1$, 47 tasks). **Middle:** Medium tasks ($0.1 \leq$ initial Success Rate $< 0.8$, 35 tasks). **Right:** Easy tasks (initial Success Rate $\geq 0.8$, 38 tasks). Shaded regions indicate standard error of the mean across tasks. The larger standard error on hard tasks reflects high variance between tasks that remain near zero throughout training and those that eventually improve.

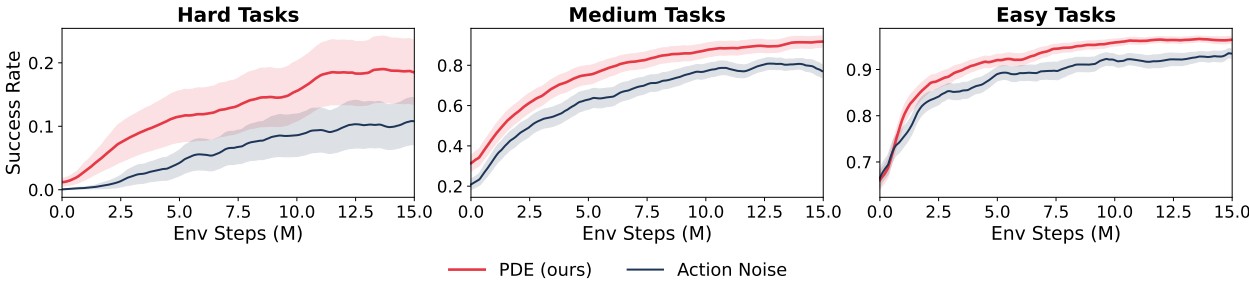

Fig. 4: Aggregated GR00T training curves grouped by task difficulty based on initial success rate. **Left:** Hard tasks (initial Success Rate $= 0$, **49** tasks). **Middle:** Medium tasks ($0 <$ initial Success Rate $\leq 0.4$, **31** tasks). **Right:** Easy tasks (initial Success Rate $> 0.4$, **40** tasks).

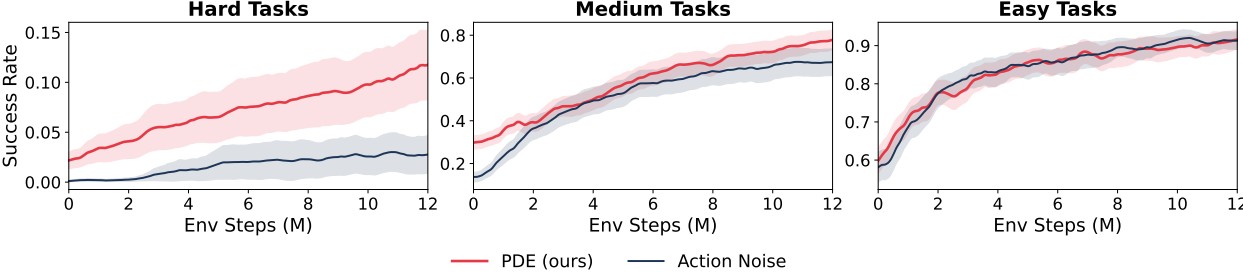

Fig. 5: Aggregated Pi0 training curves grouped by task difficulty based on initial success rate. **Left:** Hard tasks (initial Success Rate $= 0$, **49** tasks). **Middle:** Medium tasks ($0 <$ initial Success Rate $\leq 0.4$, **24** tasks). **Right:** Easy tasks (initial Success Rate $> 0.4$, **27** tasks).

**Does PDE improve over action-space exploration?** We aggregate results in two ways: by initial difficulty, defined by the initial policy's success rate (*hard*: $< 25\%$, *medium*: $25$–$50\%$, *easy*: $\geq 50\%$), and by generalization scenario. Table I reports both views. PDE outperforms Action Noise in every difficulty tier and generalization scenario. Although Action Noise already reaches nontrivial success on ManiSkill hard tasks, PDE consistently improves performance across the full benchmark, showing that prompt-space exploration remains useful beyond settings where action-space exploration completely fails.

*E. Analysis*

Table II shows three regimes where PDE improves over standard exploration and dense-reward baselines:

**Faster learning when action noise can eventually succeed.** In Regime 1, Action Noise eventually reaches high success ($93.5\%$ at step 120), but PDE learns much faster: it starts with nonzero success from exploratory prompts ($24.4\%$ at step 0) and reaches $99.0\%$ by step 30, compared with $65.2\%$ for Action Noise.

**Bootstrapping when action noise fails.** In Regime 2, Action Noise, Robometer, and RND remain at $0\%$ throughout training. PDE, however, obtains nonzero success from prompt-space

TABLE I: Aggregated success rate (%) on ManiSkill, by difficulty and by generalization axis.

| | **Difficulty** | | | **Generalization** | | | |
|---|---|---|---|---|---|---|---|
| | **Hard** | **Medium** | **Easy** | **Visual** | **Semantic** | **Execution** | **All** |
| Initial policy | $12.17_{\pm0.87}$ | $35.16_{\pm1.08}$ | $64.96_{\pm1.48}$ | $48.16_{\pm2.59}$ | $29.58_{\pm3.17}$ | $28.31_{\pm2.54}$ | $37.59_{\pm1.76}$ |
| Action Noise | $51.65_{\pm2.56}$ | $63.39_{\pm2.57}$ | $71.73_{\pm2.95}$ | $72.12_{\pm2.24}$ | $54.64_{\pm3.21}$ | $53.96_{\pm2.73}$ | $62.31_{\pm1.67}$ |
| PDE (Ours) | $\mathbf{58.77_{\pm2.73}}$ | $\mathbf{76.68_{\pm2.58}}$ | $\mathbf{84.33_{\pm2.14}}$ | $\mathbf{85.12_{\pm1.86}}$ | $\mathbf{63.65_{\pm3.29}}$ | $\mathbf{63.75_{\pm2.70}}$ | $\mathbf{73.32_{\pm1.64}}$ |

TABLE II: Success rate (%) on object-task perturbation for tasks 0, 5, 2 (Regimes 1–3) at steps 0, 30, 120.

| | **Regime 1** | | | **Regime 2** | | | **Regime 3** | | |
|---|---|---|---|---|---|---|---|---|---|
| | $s=0$ | $s=30$ | $s=120$ | $s=0$ | $s=30$ | $s=120$ | $s=0$ | $s=30$ | $s=120$ |
| Action Noise | 0.0 | 65.2 | 93.5 | 0.0 | 0.0 | 0.0 | 0.0 | 0.0 | 0.0 |
| Parameter Noise | 0.0 | 0.0 | 0.0 | 33.3 | 7.8 | 0.0 | 47.1 | 41.4 | 0.0 |
| Robometer | 0.0 | 42.3 | 97.6 | 0.0 | 0.0 | 0.0 | 0.0 | 0.0 | 0.0 |
| RND | 0.0 | 0.0 | 6.2 | 0.0 | 0.0 | 0.0 | 0.0 | 0.0 | 0.0 |
| PDE (Ours) | 24.4 | **99.0** | **100.0** | 4.3 | **29.5** | **77.7** | 0.0 | 0.0 | **81.8** |

exploration (4.3% at step 0) and improves to 77.7% by step 120, showing that alternative prompts can provide the successful rollouts needed for RL to begin improving.

**Transfer when prompt discovery alone is insufficient.** In Regime 3, PDE has 0% success at steps 0 and 30, indicating that prompt discovery alone does not immediately solve the task. Nevertheless, after RL training, PDE reaches 81.8% success by step 120, while Action Noise, Robometer, and RND remain at 0%. This suggests that successful exploratory rollouts from related tasks can train reusable skills that transfer to harder tasks.

*F. Ablation Studies*

**How does the number of discovered prompts scale with the search budget?** Prompt discovery is both consistent across runs and scales reliably with the search budget: the cumulative number of nonzero prompts grows approximately linearly with rollouts, with each independent run finding 9–12 nonzero prompts after 84 rollouts (Figure 6). This suggests that allocating more rollouts directly translates to a larger and more diverse curriculum pool.

**Does PDE produce a paraphrase-robust policy?** PDE trains under a learned distribution over prompts, so the resulting policy is exposed to many surface realizations of the same goal during fine-tuning. We construct two prompt sets of size 5: *(i) IN-Distribution Train Prompts*—the canonical instruction $p_g$="*close the microwave*" together with the four exploratory phrasings PDE drew from $\rho(\cdot \mid g, \mathcal{H})$ during training; *(ii) Out-Of-Distribution Paraphrases*—five LLM-generated rephrasings that the policy never saw. As shown in Table III, PDE generalizes across the surface form of the instruction, with no drop on unseen prompts.

**Is a single best prompt enough?** PDE mixes rollouts from a discovered prompt pool and gradually shifts training toward

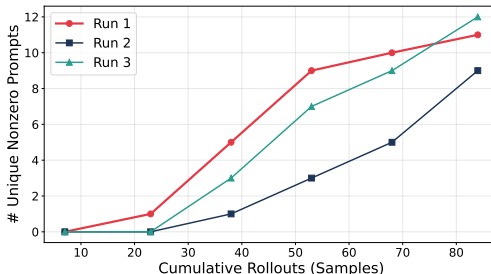

Fig. 6: Cumulative unique nonzero prompts discovered vs. cumulative rollouts across three independent runs on the microwave task.

TABLE III: Success Rate (%) on the Microwave task across 10 prompts (5 IND, 5 OOD).

| | **IND Prompts** | **OOD Prompts** |
|---|---|---|
| Action Noise | $11.2_{\pm2.9}$ | $10.4_{\pm3.0}$ |
| PDE (Ours) | $\mathbf{88.8_{\pm2.0}}$ | $\mathbf{95.2_{\pm2.9}}$ |

the canonical prompt $p_g$. A natural alternative is to train on only the single highest-success prompt, or to converge to the best discovered prompt rather than $p_g$. Evaluated under each method's own training prompt, the single-best-prompt variant scores highest (65.4% vs. 59.6% for PDE, Table V), since the policy specializes entirely on one instruction. However, this advantage disappears at deployment: when evaluated under the *original* instruction $p_g$, training on a single discovered prompt drops to 39.6% SR while PDE reaches 63.8% (Table IV)—a gap of over 24 points. The discovered prompt acts as a scaffold that unlocks early reward, but converging back to $p_g$ during training is what transfers that improvement to the deployment instruction.

TABLE IV: Eval SR on original prompt.

| Method | SR |
|---|---|
| Best Prompt (no curr.) | 0.396 |
| Prompt Curriculum | **0.638** |

TABLE V: Ablations on Object/task perturbation at step 120 (250 envs, mean across 10 tasks). Default variant shaded.

| Ablation | Variant | SR (%) |
|---|---|---|
| Training prompts | Single best prompt | **65.4** |
| | Converge to Best | 63.7 |
| | Full pool (Ours) | 59.6 |
| Backpropagation | Original-only | 49.6 |
| | Curriculum-only | 45.0 |
| | Mixed (Ours) | **59.6** |
| Consolidation $c$ | $c=0.25$ | 58.8 |
| | $c=0.5$ (Ours) | 59.6 |
| | $c=0.75$ | **63.7** |

**Does mixed backpropagation help?** Backpropagating on the original prompt alone receives sparse reward early in training and learns slowly; backpropagating on the curriculum prompt alone learns faster but plateaus, since gradient updates are never conditioned on the deployment prompt. Mixed backpropagation achieves the highest final success rate by combining the dense gradient signal of the curriculum with direct optimization on the deployment instruction (Table V).

**How sensitive is the curriculum to the consolidation target $c$?** Table V shows that all three settings ($c \in \{0.25, 0.5, 0.75\}$) converge to comparable final success rates within per-task SE, indicating that the method is not sensitive to this hyperparameter. We use $c=0.5$ as the default.

### G. Real-World Manipulation

The simulation results in Section V-C show that prompt-space exploration is most valuable precisely when the base policy receives little or no reward signal—the regime that dominates real-world RL, where rollouts are expensive and zero-reward training runs are prohibitively wasteful. We test whether PDE delivers the same advantage on a physical platform.

Our hardware setup consists of a Franka FR3 arm, a set of manipulable objects, two drawers, and a two-level rack. We first perform supervised fine-tuning of $\pi_{0.5}$ on 10 language-conditioned tasks spanning a range of difficulty and horizon lengths, from short-horizon pick-and-place to multi-stage manipulation. Starting from this SFT checkpoint, we run offline prompt optimization with a VLM supervisor on a subset of tasks where the checkpoint achieves low success, then fine-tune the policy with PDE using on-robot rollouts under the discovered prompt curriculum. We then do real-world RL following the method detailed in section IV. We find that PDE improves average task success from $20\%$ (SFT checkpoint) to $55\%$ in 10 PPO updates, corresponding to roughly 3.5 hours of on-robot training, while the action noise baseline reaches

only $25\%$ success rate.

### H. Applications to Language Model Tasks

We also show the generality of PDE beyond robotics by applying it to language-model RL tasks: competitive programming on LiveCodeBench [19] and mathematical reasoning on AIME 2026 [11]. In both settings, each problem $g$ has a canonical prompt $p_g$, PDE samples an alternative prompt $p \sim \rho(\cdot \mid g, \mathcal{H})$ during training, and evaluation uses only $p_g$. Full details and learning curves are provided in Appendix G.

**LiveCodeBench.** We evaluate PDE with RLOO [2] using 600 training problems and held-out evaluation under canonical prompts. PDE improves early sample efficiency: PDE+RLOO reaches $50\%$ held-out accuracy by the fourth training step, while RLOO reaches this level at the eighth step. PDE+RLOO also achieves larger early gains, including $42.4\%$ vs. $29.9\%$ at step 1, $53.1\%$ vs. $36.0\%$ at step 3, and $54.6\%$ vs. $41.3\%$ at step 4.

**AIME 2026.** We fine-tune Qwen3-4B [51] with GRPO [37] on all 30 AIME 2026 problems. PDE matches GRPO in final accuracy while modestly improving early learning: after the first iteration, GRPO+PDE reaches $48.9\%$ accuracy compared to $45.8\%$ for GRPO, and both methods reach $53.3\%$ after three iterations.

Together, these results show that prompt-space exploration is not specific to VLA control: it can also expose useful solution modes in language-model RL and improve early training efficiency under the same optimization budget.

## VI. CONCLUSION

We introduced Prompt-Driven Exploration (PDE), which performs posterior sampling over prompts to explore globally different behaviors. By updating a prompt distribution from rollout feedback, PDE improves sample efficiency on LIBERO and LIBERO-PRO, solves tasks where action-space exploration fails, and extends to language-model RL. More broadly, PDE shows that for foundation-model policies, effective exploration can come from the context that conditions the policy, not only from perturbing its actions.

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

*A. VLM Prompt Sampler as an Implicit Posterior*

PDE represents the prompt distribution $\rho(\cdot \mid g, \mathcal{H})$ with a VLM supervisor $\mathcal{V}$. As in Section IV-A, the interaction history before iteration $t$ is

$$\mathcal{H}_t = \{(g_i, p_i, \tau_i, R(\tau_i, g_i))\}_{i<t}, \tag{4}$$

where $g_i$ is the task, $p_i$ is the prompt used for rollout, $\tau_i$ is the resulting trajectory, and $R(\tau_i, g_i)$ is the task reward.

In implementation, we store a compact task-specific history to fit the VLM context window. For each task $g$, we define

$$\widetilde{\mathcal{H}}_t^g = \{(p_i, c_i, s_i) \ : \ g_i = g, \ i < t\}, \tag{5}$$

where $s_i \in [0, 1]$ is the empirical success rate of prompt $p_i$ over $N$ rollouts, and $c_i$ is a one-sentence VLM summary of the corresponding rollout behavior. Thus, $\widetilde{\mathcal{H}}_t^g$ is a compressed representation of the full history $\mathcal{H}_t$ restricted to task $g$.

We also maintain a task-specific prompt pool

$$\mathcal{P}_t^g \subseteq \{p_i : (p_i, c_i, s_i) \in \widetilde{\mathcal{H}}_t^g\}, \tag{6}$$

containing prompts whose empirical success exceeds a threshold $\eta$. At prompt-update iteration $t$, we query the VLM as

$$p \sim \rho_t(\cdot \mid g, \mathcal{H}_t) \ \approx \ \mathcal{V}(\cdot \mid p_g, \mathcal{P}_t^g, \widetilde{\mathcal{H}}_t^g), \tag{7}$$

where $p_g$ is the canonical prompt for task $g$. Sampling from the prompt posterior therefore reduces to asking the VLM for candidate prompts conditioned on the canonical prompt, the successful prompt pool, and compressed rollout feedback. Updating the posterior is implemented by appending the latest rollout summaries and success rates to $\widetilde{\mathcal{H}}_t^g$ and re-querying the VLM.

*a) Rollout summaries.:* After evaluating a prompt for $N$ rollouts, we query $\mathcal{V}$ once on the rollout videos and ask for a single-sentence description of what the policy attempted and where it failed. We store only this sentence $c_i$, not the videos, so the context grows with the number of evaluated prompts rather than the number of frames. The exact summary prompt is given in Appendix E.

*b) Pool admission.:* At each prompt-update iteration, we evaluate $K$ candidate prompts. A candidate $p$ is added to $\mathcal{P}_t^g$ if its empirical success rate satisfies $s \geq \eta$. Prompts with $s < \eta$ are still stored in $\widetilde{\mathcal{H}}_t^g$ as negative feedback. Prompts with perfect success are kept in the pool without further refinement.

*B. Prompt Discovery and Policy Optimization Schedule*

In the VLA experiments, we separate prompt discovery from policy optimization so that the prompt sampler is updated against a fixed initial policy. This avoids a moving target: if $\theta$ changes while prompts are being evaluated, the same prompt may appear good or bad for reasons unrelated to the prompt update.

We use a two-stage schedule:

- **Prompt-posterior update.** We freeze $\theta$. For each task $g$, PDE samples candidate prompts from $\rho_t(\cdot \mid g, \mathcal{H}_t)$, evaluates each prompt for $N$ rollouts, and updates $\widetilde{\mathcal{H}}_t^g$ and $\mathcal{P}_t^g$.
- **RL update.** We freeze the discovered prompt pools and optimize $\theta$ with PPO. Rollouts are collected from a mixture of the canonical prompt $p_g$ and the discovered prompt pool $\mathcal{P}^g$, as described below.

We enter the RL stage once the prompt pool contains at least $M_{\min}$ prompts, or after a maximum prompt-search budget is reached.

*C. Anchoring Policy Updates to the Canonical Prompt*

During RL, exploratory prompts provide successful rollouts, but evaluation is always under the canonical prompt $p_g$. We therefore use two mechanisms to transfer improvements from exploratory prompts back to $p_g$.

*a) Mixture sampling.:* For task $g$, each rollout prompt is sampled as

$$p \sim \alpha_t \delta_{p_g} + (1 - \alpha_t)\mathrm{Uniform}(\mathcal{P}^g), \tag{8}$$

where $\mathcal{P}^g$ is the frozen prompt pool found during prompt discovery. Prompts from $\mathcal{P}^g$ provide reward signal early in training, while rollouts under $p_g$ keep optimization aligned with evaluation.

TABLE VI: Hyperparameters for PDE. Values in the top two blocks are specific to PDE; the bottom block lists the standard PPO hyperparameters used for policy-gradient updates in Regime B.

| Symbol | Description | Value |
|---|---|---|
| *Posterior-sampling loop* | | |
| $T_0$ | Pool-building iterations | 10 |
| $K$ | Candidate prompts per iteration | 5 |
| $N$ | Rollouts per candidate | 10 |
| $\eta$ | Pool admission threshold | $> 0$ |
| *Deployment-prompt anchoring* | | |
| $\alpha_{\min}$ | Floor on deployment-prompt mixture weight | 0.05 |
| $\beta$ | EMA smoothing factor | 0.3 |
| $c$ | Consolidation target | 0.5 |
| *PPO* | | |
| $\epsilon$ | Clip range | 0.2 |
| $\eta_{\mathrm{lr}}$ | Learning rate | $5 \times 10^{-6}$ |
| $B$ | Batch size | 2048 |
| $\lambda$ | GAE $\lambda$ | 0.95 |
| $\gamma$ | Discount factor | 0.99 |
| $E$ | Epochs per update | 4 |

*b) Adaptive mixture coefficient.:* Let $s_t^{(p_g)}$ be the empirical success rate of rollouts collected under the canonical prompt at training step $t$. We track an exponential moving average

$$\bar{s}_t = \beta s_t^{(p_g)} + (1 - \beta)\bar{s}_{t-1}, \tag{9}$$

$$\alpha_t = \mathrm{clip}\left(\frac{\bar{s}_t}{c}, \alpha_{\min}, 1\right), \tag{10}$$

where $\beta$ is the smoothing factor, $c$ is the consolidation target, and $\alpha_{\min} > 0$ ensures that the canonical prompt is always sampled. When canonical-prompt success is low, most rollouts come from the discovered prompt pool. As canonical-prompt success improves, $\alpha_t$ increases and training shifts toward $p_g$.

*c) Mixed backpropagation.:* A rollout collected under an exploratory prompt $p$ directly updates $\pi_\theta(\cdot \mid o, p)$, but evaluation uses $\pi_\theta(\cdot \mid o, p_g)$. To couple these two prompt-conditioned policies, we replace the current-policy log-probability in the PPO ratio with

$$\ell_t(\theta) = \frac{1}{2} \log \pi_\theta(a_t \mid o_t, p_g) + \frac{1}{2} \log \pi_\theta(a_t \mid o_t, p). \tag{11}$$

The PPO ratio becomes

$$w_t(\theta) = \exp(\ell_t(\theta) - \log \pi_{\mathrm{old}}(a_t \mid o_t, p)), \tag{12}$$

where the denominator uses the old log-probability recorded under the rollout prompt $p$. Gradients flow through both forward passes. This encourages actions that are likely under both the exploratory prompt and the canonical prompt, helping transfer successful exploratory behavior back to the policy evaluated under $p_g$.

*D. Hyperparameters*

Table VI lists all hyperparameters used in our experiments. Unless stated otherwise, values are shared across LIBERO and LIBERO-PRO.

*E. Supervisor Prompt Templates*

We reproduce the three prompt templates used to query $\mathcal{V}$: (i) the candidate-generation template, which takes $(p_0^g, \mathcal{P}_t, \mathcal{H}_t)$ as input and returns $K$ proposed prompts; (ii) the rollout-summarization template, which takes $N$ rollout videos of a single prompt as input and returns a one-sentence summary $c$; (iii) the system prompt that frames the supervisor's role. Each template is given verbatim below, with placeholders in `{curly braces}`.

TABLE VII: Pi0.5 aggregated success rate (%) on LIBERO-PRO, by task difficulty and by perturbation axis.

| | Difficulty | | | Perturbation | | | All |
|---|---|---|---|---|---|---|---|
| | **Hard** | **Medium** | **Easy** | **Object** | **Swap** | **Task** | **All** |
| Initial policy | $1.28\pm0.40$ | $45.08\pm3.64$ | $94.32\pm0.88$ | $61.26\pm5.72$ | $50.31\pm6.55$ | $18.99\pm5.24$ | $43.52\pm3.73$ |
| Action Noise | $13.32\pm4.30$ | $78.16\pm2.88$ | $95.97\pm1.06$ | $71.97\pm5.46$ | $67.33\pm6.10$ | $35.91\pm7.11$ | $58.40\pm3.87$ |
| Action Noise (n=0.7) | $17.00\pm4.82$ | $74.27\pm3.65$ | $90.53\pm1.93$ | $71.96\pm5.46$ | $64.74\pm5.89$ | $34.26\pm6.56$ | $56.99\pm3.74$ |
| Parameter Noise (n=0.01) | $13.03\pm4.32$ | $77.54\pm3.26$ | $96.11\pm0.64$ | $72.89\pm5.52$ | $67.25\pm6.16$ | $34.32\pm6.97$ | $58.15\pm3.90$ |
| Parameter Noise (n=0.1) | $0.77\pm0.52$ | $2.36\pm1.24$ | $18.40\pm3.48$ | $9.68\pm2.83$ | $8.39\pm2.18$ | $2.38\pm1.93$ | $6.82\pm1.37$ |
| Flow Noise + Ent | $17.24\pm4.80$ | $81.08\pm3.05$ | $95.42\pm0.99$ | $75.61\pm5.43$ | $68.04\pm6.07$ | $38.20\pm7.01$ | $60.62\pm3.85$ |
| VLAC | $7.07\pm2.89$ | $63.59\pm4.95$ | $93.26\pm1.67$ | $68.20\pm5.73$ | $61.75\pm6.34$ | $22.60\pm5.81$ | $50.85\pm3.88$ |
| Robometer | $13.34\pm4.27$ | $77.40\pm3.48$ | $95.79\pm1.50$ | $73.33\pm5.44$ | $66.24\pm6.30$ | $34.83\pm7.00$ | $58.13\pm3.91$ |
| RND | $10.53\pm3.90$ | $77.53\pm2.95$ | $90.93\pm2.31$ | $69.88\pm5.37$ | $66.17\pm6.10$ | $30.54\pm6.68$ | $55.53\pm3.84$ |
| PDE (Ours) | $\mathbf{19.48\pm5.10}$ | $\mathbf{85.15\pm2.58}$ | $\mathbf{97.46\pm0.86}$ | $\mathbf{77.31\pm5.50}$ | $\mathbf{71.24\pm6.17}$ | $\mathbf{41.43\pm7.17}$ | $\mathbf{63.33\pm3.89}$ |

*a) (i) Candidate generation and rollout summarization:*

```
## Training Step {step} -- Prompt Pool
**GOAL: {task_description}**
**Original prompt**: "{task_description}" | EMA success: {orig_rate} [window trend:
{trend}]
### Current Pool Prompts and Their Rollout Videos
**Prompt**: "{prompt_i}" | EMA success: {ema_i}
{rollout video or frames for prompt_i}
... (repeated for each prompt in the pool)
### Your Task
1. **Analyze each video**: What is the robot doing under each prompt? Which prompts
produce the best behavior toward the goal?
2. **Summarize each prompt's effect** in one line.
3. **Generate 1 NEW prompt** to add to the pool:
- Address failure modes you observed
- Be semantically diverse from existing prompts
- Try different phrasings, action verbs, spatial references
Return ONLY valid JSON:
{
"new_prompts": ["new prompt"],
"summaries": {"exact prompt text": "one-line behavior summary", ...},
"analysis": "brief reasoning about what to try next"
}
```

*b) (ii) System prompt:*

```
You are an expert in robotic manipulation and prompt engineering for
vision-language-action models.
Each prompt is accompanied by {frames_per_video} frames from a robot rollout video,
shown in temporal order (Frame 1 = start, Frame {frames_per_video} = end). Trace the
robot's movement across frames to understand what it does over time.
Your tasks:
1. Analyze the rollout videos -- what is the robot doing? How close to the goal?
2. Identify failure modes (wrong object, wrong location, failed grasp, etc.)
3. Analyze how different prompts affect the robot's behavior
4. Generate 1 NEW prompt to add to the pool
The GOAL is: "{task_description}"
The robot is controlled by a vision-language-action model (pi0.5).
Guidelines for new prompts:
- Be clear and specific about the object and target location
- Use natural language similar to LIBERO benchmark conventions
- Try different levels of specificity, action verbs, spatial references
- Keep prompts concise (5-15 words typically)
```

## F. Compute

For each experiment, we used one node (8) of H200s.

## G. Detailed results

TABLE VIII: GR00T and Pi0 aggregated success rate (%) on LIBERO-PRO by task difficulty.

| | GR00T | | | | Pi0 | | | |
|---|---|---|---|---|---|---|---|---|
| | **Hard** | **Medium** | **Easy** | **All** | **Hard** | **Medium** | **Easy** | **All** |
| Action Noise | 12.73±4.24 | 87.23±1.90 | 96.00±0.86 | 59.73±4.01 | 12.16±4.03 | 73.83±7.05 | 88.44±4.83 | 47.56±4.56 |
| PDE (Ours) | **20.33**±5.49 | **93.68**±3.16 | **98.30**±0.42 | **65.27**±4.17 | **12.73**±3.77 | **85.00**±4.50 | **89.33**±4.75 | **50.76**±4.49 |

TABLE IX: Per-task success rate (%) on LIBERO-PRO with GR00T, sorted by difficulty bucket and PPO init SR.

| Diff. | Suite | Pert. | Task | Init | PPO | Ours |
|---|---|---|---|---|---|---|
| easy | goal | object | 7 | 100.00 | 96.00 | **96.00** |
| easy | 10 | swap | 1 | 83.67 | 96.00 | **96.00** |
| easy | object | swap | 6 | 79.25 | 100.00 | **92.00** |
| easy | 10 | swap | 5 | 78.43 | 96.00 | **92.00** |
| easy | object | object | 8 | 78.33 | 96.00 | **100.00** |
| easy | object | swap | 8 | 78.33 | 100.00 | **100.00** |
| easy | 10 | object | 1 | 77.55 | 92.00 | **100.00** |
| easy | object | object | 4 | 76.79 | 96.00 | **100.00** |
| easy | 10 | object | 3 | 75.56 | 96.00 | **100.00** |
| easy | object | swap | 3 | 75.56 | 100.00 | **100.00** |
| easy | object | swap | 4 | 75.00 | 100.00 | **100.00** |
| easy | spatial | swap | 1 | 69.39 | 96.00 | **96.00** |
| easy | 10 | swap | 6 | 67.92 | 100.00 | **100.00** |
| easy | object | object | 6 | 67.92 | 100.00 | **100.00** |
| easy | 10 | object | 4 | 67.86 | 80.00 | **96.00** |
| easy | object | swap | 5 | 66.67 | 100.00 | **100.00** |
| easy | object | object | 1 | 63.27 | 100.00 | **100.00** |
| easy | object | swap | 1 | 63.27 | 100.00 | **100.00** |
| easy | object | object | 7 | 62.79 | 100.00 | **100.00** |
| easy | 10 | object | 2 | 62.75 | 96.00 | **100.00** |
| easy | 10 | swap | 4 | 60.71 | 88.00 | **96.00** |
| easy | object | task | 1 | 59.18 | 100.00 | **100.00** |
| easy | 10 | swap | 2 | 58.82 | 96.00 | **100.00** |
| easy | spatial | swap | 5 | 58.82 | 84.00 | **100.00** |
| easy | spatial | object | 1 | 57.14 | 84.00 | **100.00** |
| easy | object | object | 9 | 56.36 | 100.00 | **100.00** |
| easy | object | swap | 7 | 55.81 | 100.00 | **96.00** |
| easy | spatial | swap | 0 | 55.10 | 100.00 | **100.00** |
| easy | spatial | swap | 2 | 54.90 | 84.00 | **92.00** |
| easy | object | swap | 0 | 53.06 | 96.00 | **100.00** |
| easy | spatial | object | 0 | 53.06 | 100.00 | **96.00** |
| easy | spatial | object | 2 | 50.98 | 88.00 | **100.00** |
| easy | goal | swap | 5 | 49.02 | 92.00 | **100.00** |
| easy | 10 | swap | 0 | 48.98 | 92.00 | **92.00** |
| easy | object | object | 3 | 46.67 | 100.00 | **100.00** |
| easy | 10 | swap | 7 | 46.51 | 100.00 | **100.00** |
| easy | object | swap | 9 | 45.45 | 100.00 | **96.00** |
| easy | object | object | 5 | 45.10 | 100.00 | **100.00** |
| easy | goal | task | 7 | 44.19 | 100.00 | **100.00** |
| easy | goal | object | 5 | 41.18 | 96.00 | **96.00** |
| medium | goal | swap | 8 | 38.33 | 88.00 | **88.00** |
| medium | spatial | object | 5 | 35.29 | 92.00 | **100.00** |
| medium | spatial | object | 6 | 33.96 | 96.00 | **100.00** |

*(continued on next page)*

| Diff. | Suite | Pert. | Task | Init | PPO | EMA (Ours) |
|---|---|---|---|---|---|---|
| medium | spatial | object | 3 | 33.33 | 100.00 | **100.00** |
| medium | spatial | swap | 9 | 30.91 | 80.00 | **100.00** |
| medium | object | object | 2 | 29.41 | 100.00 | **100.00** |
| medium | spatial | object | 8 | 28.33 | 88.00 | **96.00** |
| medium | spatial | swap | 8 | 28.33 | 88.00 | **96.00** |
| medium | spatial | swap | 6 | 28.30 | 92.00 | **100.00** |
| medium | goal | object | 2 | 27.45 | 64.00 | **100.00** |
| medium | 10 | object | 9 | 25.45 | 64.00 | **0.00** |
| medium | spatial | object | 9 | 23.64 | 84.00 | **84.00** |
| medium | spatial | swap | 7 | 23.26 | 64.00 | **96.00** |
| medium | spatial | task | 5 | 21.57 | 92.00 | **92.00** |
| medium | spatial | task | 9 | 20.00 | 96.00 | **100.00** |
| medium | spatial | object | 4 | 19.64 | 96.00 | **96.00** |
| medium | object | swap | 2 | 19.61 | 96.00 | **100.00** |
| medium | spatial | swap | 4 | 16.07 | 96.00 | **100.00** |
| medium | 10 | object | 8 | 13.33 | 68.00 | **96.00** |
| medium | goal | object | 9 | 12.73 | 88.00 | **92.00** |
| medium | spatial | object | 7 | 11.63 | 80.00 | **92.00** |
| medium | spatial | task | 7 | 11.63 | 84.00 | **96.00** |
| medium | goal | object | 4 | 10.71 | 92.00 | **100.00** |
| medium | goal | object | 0 | 10.20 | 96.00 | **100.00** |
| medium | spatial | task | 2 | 7.84 | 92.00 | **100.00** |
| medium | 10 | task | 0 | 6.12 | 84.00 | **92.00** |
| medium | 10 | task | 8 | 5.00 | 80.00 | **96.00** |
| medium | goal | object | 1 | 4.08 | 100.00 | **100.00** |
| medium | object | object | 0 | 2.04 | 76.00 | **100.00** |
| medium | 10 | object | 5 | 1.96 | 96.00 | **100.00** |
| medium | goal | task | 5 | 1.96 | 92.00 | **92.00** |
| hard | 10 | object | 0 | 0.00 | 0.00 | **0.00** |
| hard | 10 | object | 6 | 0.00 | 92.00 | **100.00** |
| hard | 10 | object | 7 | 0.00 | 0.00 | **0.00** |
| hard | 10 | swap | 3 | 0.00 | 0.00 | **0.00** |
| hard | 10 | swap | 8 | 0.00 | 0.00 | **0.00** |
| hard | 10 | swap | 9 | 0.00 | 0.00 | **0.00** |
| hard | 10 | task | 1 | 0.00 | 0.00 | **0.00** |
| hard | 10 | task | 2 | 0.00 | 0.00 | **0.00** |
| hard | 10 | task | 3 | 0.00 | 0.00 | **0.00** |
| hard | 10 | task | 4 | 0.00 | 0.00 | **0.00** |
| hard | 10 | task | 5 | 0.00 | 0.00 | **0.00** |
| hard | 10 | task | 6 | 0.00 | 0.00 | **0.00** |
| hard | 10 | task | 7 | 0.00 | 0.00 | **0.00** |
| hard | 10 | task | 9 | 0.00 | 0.00 | **0.00** |
| hard | goal | object | 3 | 0.00 | 0.00 | **0.00** |
| hard | goal | object | 6 | 0.00 | 36.00 | **48.00** |
| hard | goal | object | 8 | 0.00 | 80.00 | **92.00** |
| hard | goal | swap | 0 | 0.00 | 0.00 | **0.00** |
| hard | goal | swap | 1 | 0.00 | 0.00 | **0.00** |
| hard | goal | swap | 2 | 0.00 | 0.00 | **96.00** |
| hard | goal | swap | 3 | 0.00 | 88.00 | **88.00** |
| hard | goal | swap | 4 | 0.00 | 0.00 | **0.00** |
| hard | goal | swap | 6 | 0.00 | 0.00 | **0.00** |

| Diff. | Suite | Pert. | Task | Init | PPO | EMA (Ours) |
|-------|-------|-------|------|------|-----|------------|
| hard | goal | swap | 7 | 0.00 | 0.00 | **100.00** |
| hard | goal | swap | 9 | 0.00 | 0.00 | **100.00** |
| hard | goal | task | 0 | 0.00 | 0.00 | **100.00** |
| hard | goal | task | 1 | 0.00 | 0.00 | **0.00** |
| hard | goal | task | 2 | 0.00 | 0.00 | **0.00** |
| hard | goal | task | 3 | 0.00 | 0.00 | **0.00** |
| hard | goal | task | 4 | 0.00 | 0.00 | **0.00** |
| hard | goal | task | 6 | 0.00 | 0.00 | **0.00** |
| hard | goal | task | 8 | 0.00 | 4.00 | **0.00** |
| hard | goal | task | 9 | 0.00 | 0.00 | **0.00** |
| hard | object | task | 0 | 0.00 | 0.00 | **0.00** |
| hard | object | task | 2 | 0.00 | 0.00 | **0.00** |
| hard | object | task | 3 | 0.00 | 0.00 | **0.00** |
| hard | object | task | 4 | 0.00 | 0.00 | **0.00** |
| hard | object | task | 5 | 0.00 | 0.00 | **0.00** |
| hard | object | task | 6 | 0.00 | 0.00 | **0.00** |
| hard | object | task | 7 | 0.00 | 0.00 | **0.00** |
| hard | object | task | 8 | 0.00 | 0.00 | **0.00** |
| hard | object | task | 9 | 0.00 | 0.00 | **0.00** |
| hard | spatial | swap | 3 | 0.00 | 0.00 | **0.00** |
| hard | spatial | task | 0 | 0.00 | 0.00 | **80.00** |
| hard | spatial | task | 1 | 0.00 | 0.00 | **0.00** |
| hard | spatial | task | 3 | 0.00 | 92.00 | **0.00** |
| hard | spatial | task | 4 | 0.00 | 56.00 | **0.00** |
| hard | spatial | task | 6 | 0.00 | 96.00 | **100.00** |
| hard | spatial | task | 8 | 0.00 | 80.00 | **92.00** |

TABLE X: Per-task success rate (%) on LIBERO-PRO with Pi0, sorted by difficulty bucket and PPO init SR.

| Diff. | Suite | Pert. | Task | Init | PPO | Ours |
|-------|-------|-------|------|------|-----|------|
| easy | goal | object | 1 | 93.88 | 100.00 | **100.00** |
| easy | object | swap | 4 | 89.29 | 96.00 | **100.00** |
| easy | goal | swap | 8 | 86.67 | 100.00 | **100.00** |
| easy | spatial | object | 2 | 86.27 | 100.00 | **100.00** |
| easy | spatial | swap | 2 | 82.35 | 100.00 | **100.00** |
| easy | spatial | object | 0 | 77.55 | 8.00 | **96.00** |
| easy | spatial | swap | 0 | 77.55 | 92.00 | **88.00** |
| easy | goal | object | 5 | 74.51 | 0.00 | **96.00** |
| easy | spatial | object | 3 | 66.67 | 96.00 | **100.00** |
| easy | object | object | 4 | 62.50 | 100.00 | **100.00** |
| easy | object | swap | 8 | 60.00 | 100.00 | **100.00** |
| easy | spatial | task | 5 | 56.86 | 100.00 | **96.00** |
| easy | object | swap | 9 | 56.36 | 92.00 | **92.00** |
| easy | object | swap | 0 | 55.10 | 100.00 | **100.00** |
| easy | object | swap | 2 | 54.90 | 96.00 | **96.00** |
| easy | goal | object | 8 | 53.33 | 80.00 | **0.00** |
| easy | goal | swap | 5 | 50.98 | 100.00 | **96.00** |
| easy | object | swap | 5 | 47.06 | 96.00 | **96.00** |
| easy | goal | object | 7 | 46.51 | 100.00 | **100.00** |
| easy | spatial | object | 4 | 44.64 | 80.00 | **12.00** |

| Diff. | Suite | Pert. | Task | Init | PPO | EMA (Ours) |
|---|---|---|---|---|---|---|
| easy | goal | task | 7 | 44.19 | 68.00 | **92.00** |
| easy | object | swap | 6 | 43.40 | 100.00 | **100.00** |
| easy | object | object | 5 | 43.14 | 96.00 | **92.00** |
| easy | spatial | swap | 4 | 41.07 | 88.00 | **64.00** |
| easy | object | object | 1 | 40.82 | 100.00 | **100.00** |
| easy | object | swap | 1 | 40.82 | 100.00 | **96.00** |
| easy | object | object | 8 | 40.00 | 100.00 | **100.00** |
| medium | object | object | 6 | 39.62 | 96.00 | **100.00** |
| medium | spatial | task | 1 | 32.65 | 88.00 | **20.00** |
| medium | goal | object | 9 | 27.27 | 44.00 | **72.00** |
| medium | spatial | swap | 8 | 16.67 | 72.00 | **88.00** |
| medium | spatial | swap | 1 | 16.33 | 100.00 | **92.00** |
| medium | spatial | object | 1 | 14.29 | 100.00 | **100.00** |
| medium | object | swap | 7 | 13.95 | 96.00 | **92.00** |
| medium | spatial | object | 8 | 13.33 | 88.00 | **88.00** |
| medium | object | object | 9 | 12.73 | 96.00 | **96.00** |
| medium | object | object | 0 | 12.24 | 100.00 | **100.00** |
| medium | spatial | swap | 6 | 11.32 | 100.00 | **76.00** |
| medium | goal | swap | 3 | 11.11 | 100.00 | **96.00** |
| medium | spatial | object | 9 | 10.91 | 96.00 | **92.00** |
| medium | object | object | 7 | 9.30 | 100.00 | **100.00** |
| medium | goal | swap | 7 | 6.98 | 0.00 | **96.00** |
| medium | object | swap | 3 | 6.67 | 92.00 | **100.00** |
| medium | spatial | task | 8 | 6.67 | 92.00 | **100.00** |
| medium | spatial | task | 2 | 5.88 | 88.00 | **84.00** |
| medium | spatial | object | 6 | 5.66 | 68.00 | **96.00** |
| medium | spatial | swap | 9 | 5.45 | 88.00 | **84.00** |
| medium | goal | object | 3 | 4.44 | 16.00 | **72.00** |
| medium | goal | task | 5 | 3.92 | 52.00 | **80.00** |
| medium | object | object | 2 | 3.92 | 0.00 | **100.00** |
| medium | 10 | object | 1 | 2.04 | 0.00 | **16.00** |
| hard | 10 | object | 0 | 0.00 | 0.00 | **0.00** |
| hard | 10 | object | 2 | 0.00 | 0.00 | **0.00** |
| hard | 10 | object | 3 | 0.00 | 88.00 | **84.00** |
| hard | 10 | object | 4 | 0.00 | 0.00 | **16.00** |
| hard | 10 | object | 5 | 0.00 | 0.00 | **0.00** |
| hard | 10 | object | 6 | 0.00 | 0.00 | **0.00** |
| hard | 10 | object | 7 | 0.00 | 0.00 | **0.00** |
| hard | 10 | object | 8 | 0.00 | 0.00 | **0.00** |
| hard | 10 | object | 9 | 0.00 | 0.00 | **0.00** |
| hard | 10 | swap | 0 | 0.00 | 0.00 | **0.00** |
| hard | 10 | swap | 1 | 0.00 | 0.00 | **56.00** |
| hard | 10 | swap | 2 | 0.00 | 20.00 | **0.00** |
| hard | 10 | swap | 3 | 0.00 | 0.00 | **0.00** |
| hard | 10 | swap | 4 | 0.00 | 12.00 | **0.00** |
| hard | 10 | swap | 5 | 0.00 | 0.00 | **0.00** |
| hard | 10 | swap | 6 | 0.00 | 0.00 | **0.00** |
| hard | 10 | swap | 7 | 0.00 | 0.00 | **0.00** |
| hard | 10 | swap | 8 | 0.00 | 0.00 | **0.00** |
| hard | 10 | swap | 9 | 0.00 | 0.00 | **0.00** |
| hard | goal | object | 0 | 0.00 | 20.00 | **0.00** |

| Diff. | Suite | Pert. | Task | Init | PPO | EMA (Ours) |
|-------|-------|-------|------|------|------|------------|
| hard | goal | object | 2 | 0.00 | 0.00 | **0.00** |
| hard | goal | object | 4 | 0.00 | 24.00 | **0.00** |
| hard | goal | object | 6 | 0.00 | 0.00 | **0.00** |
| hard | goal | swap | 0 | 0.00 | 0.00 | **8.00** |
| hard | goal | swap | 1 | 0.00 | 100.00 | **0.00** |
| hard | goal | swap | 2 | 0.00 | 0.00 | **0.00** |
| hard | goal | swap | 4 | 0.00 | 0.00 | **0.00** |
| hard | goal | swap | 6 | 0.00 | 0.00 | **0.00** |
| hard | goal | swap | 9 | 0.00 | 0.00 | **0.00** |
| hard | goal | task | 0 | 0.00 | 0.00 | **0.00** |
| hard | goal | task | 1 | 0.00 | 100.00 | **0.00** |
| hard | goal | task | 2 | 0.00 | 0.00 | **0.00** |
| hard | goal | task | 3 | 0.00 | 0.00 | **0.00** |
| hard | goal | task | 4 | 0.00 | 0.00 | **0.00** |
| hard | goal | task | 6 | 0.00 | 0.00 | **0.00** |
| hard | goal | task | 8 | 0.00 | 0.00 | **0.00** |
| hard | goal | task | 9 | 0.00 | 0.00 | **0.00** |
| hard | object | object | 3 | 0.00 | 92.00 | **96.00** |
| hard | spatial | object | 5 | 0.00 | 0.00 | **40.00** |
| hard | spatial | object | 7 | 0.00 | 0.00 | **0.00** |
| hard | spatial | swap | 3 | 0.00 | 4.00 | **0.00** |
| hard | spatial | swap | 5 | 0.00 | 52.00 | **20.00** |
| hard | spatial | swap | 7 | 0.00 | 4.00 | **12.00** |
| hard | spatial | task | 0 | 0.00 | 0.00 | **0.00** |
| hard | spatial | task | 3 | 0.00 | 0.00 | **72.00** |
| hard | spatial | task | 4 | 0.00 | 76.00 | **20.00** |
| hard | spatial | task | 6 | 0.00 | 0.00 | **40.00** |
| hard | spatial | task | 7 | 0.00 | 4.00 | **92.00** |
| hard | spatial | task | 9 | 0.00 | 0.00 | **68.00** |

TABLE XI: Per-task success rate (%) on ManiSkill at $\tau=1.0$, sorted by difficulty bucket and SFT init SR.

| Difficulty | Category | Variant | Task | SFT | PPO | Ours |
|------------|----------|---------|------|-----|-----|------|
| hard | semantic | MultiPlate | 0 | 23.10 | 30.80 | **33.30** |
| hard | semantic | MultiPlate | 13 | 23.10 | 30.80 | **25.00** |
| hard | semantic | MultiPlate | 11 | 22.20 | 33.30 | **66.70** |
| hard | execution | Position | 1 | 21.40 | 57.10 | **71.40** |
| hard | semantic | MultiPlate | 7 | 20.00 | 20.00 | **45.00** |
| hard | execution | PositionChangeTo | 8 | 20.00 | 55.00 | **60.00** |
| hard | execution | EEPose | 3 | 20.00 | 70.00 | **60.00** |
| hard | execution | EEPose | 5 | 20.00 | 73.30 | **93.30** |
| hard | visual | VisionTexture05 | 11 | 20.00 | 53.30 | **100.00** |
| hard | visual | VisionWhole05 | 11 | 20.00 | 53.30 | **100.00** |
| hard | visual | VisionWhole05 | 14 | 20.00 | 60.00 | **70.00** |
| hard | execution | Position | 2 | 18.80 | 56.20 | **68.80** |
| hard | execution | PositionChangeTo | 4 | 18.80 | 68.80 | **62.50** |
| hard | semantic | MultiPlate | 9 | 18.20 | 45.50 | **36.40** |
| hard | visual | VisionWhole03 | 6 | 18.20 | 100.00 | **84.60** |
| hard | semantic | MultiCarrot | 24 | 17.90 | 32.10 | **35.70** |
| hard | execution | PositionChangeTo | 12 | 17.60 | 52.90 | **85.70** |

*(continued on next page)*

| Difficulty | Category | Variant | Task | SFT | PPO | Ours |
|---|---|---|---|---|---|---|
| hard | execution | Position | 12 | 16.70 | 61.10 | **61.10** |
| hard | execution | PositionChangeTo | 14 | 16.70 | 75.00 | **58.30** |
| hard | semantic | MultiCarrot | 18 | 15.80 | 52.60 | **57.90** |
| hard | execution | Position | 11 | 15.40 | 30.80 | **38.50** |
| hard | visual | VisionWhole03 | 11 | 15.40 | 53.30 | **100.00** |
| hard | semantic | MultiPlate | 5 | 15.00 | 65.00 | **60.00** |
| hard | semantic | Plate | 2 | 14.30 | 81.00 | **85.70** |
| hard | semantic | MultiPlate | 3 | 14.30 | 64.30 | **83.30** |
| hard | semantic | MultiPlate | 8 | 14.30 | 42.90 | **50.00** |
| hard | execution | PositionChangeTo | 7 | 14.30 | 42.90 | **42.90** |
| hard | visual | VisionTexture05 | 2 | 14.30 | 76.20 | **81.00** |
| hard | visual | VisionWhole05 | 2 | 14.30 | 66.70 | **61.90** |
| hard | execution | PositionChangeTo | 10 | 13.30 | 40.00 | **50.00** |
| hard | execution | PositionChangeTo | 1 | 12.50 | 62.50 | **50.00** |
| hard | visual | VisionTexture03 | 12 | 12.50 | 81.20 | **93.80** |
| hard | semantic | MultiCarrot | 21 | 11.80 | 26.50 | **44.10** |
| hard | semantic | MultiPlate | 6 | 11.80 | 29.40 | **40.00** |
| hard | semantic | MainCarrot | 21 | 9.70 | 77.40 | **67.70** |
| hard | semantic | MultiCarrot | 23 | 9.70 | 35.50 | **44.40** |
| hard | visual | VisionTexture05 | 1 | 9.50 | 42.90 | **66.70** |
| hard | visual | VisionWhole05 | 1 | 9.50 | 38.10 | **50.00** |
| hard | semantic | MultiPlate | 4 | 9.10 | 45.50 | **36.40** |
| hard | execution | EEPose | 1 | 8.30 | 50.00 | **100.00** |
| hard | execution | PositionChangeTo | 2 | 7.70 | 53.80 | **100.00** |
| hard | execution | PositionChangeTo | 6 | 7.70 | 76.90 | **100.00** |
| hard | execution | EEPose | 11 | 7.10 | 64.30 | **50.00** |
| hard | semantic | MultiPlate | 2 | 6.20 | 31.20 | **33.30** |
| hard | semantic | MultiPlate | 14 | 6.20 | 50.00 | **50.00** |
| hard | semantic | MainCarrot | 22 | 5.90 | 47.10 | **61.80** |
| hard | semantic | MultiPlate | 10 | 5.90 | 5.90 | **17.60** |
| hard | execution | PositionChangeTo | 3 | 5.90 | 35.30 | **47.10** |
| hard | execution | PositionChangeTo | 9 | 5.90 | 47.10 | **50.00** |
| hard | visual | VisionWhole03 | 1 | 5.00 | 61.90 | **66.70** |
| hard | semantic | Plate | 1 | 4.80 | 52.40 | **57.10** |
| hard | visual | VisionTexture03 | 1 | 4.80 | 38.10 | **66.70** |
| hard | visual | VisionTexture03 | 2 | 4.80 | 100.00 | **100.00** |
| hard | semantic | MultiCarrot | 22 | 2.60 | 59.00 | **64.30** |
| hard | semantic | MultiPlate | 1 | 0.00 | 28.60 | **40.00** |
| hard | semantic | MultiPlate | 12 | 0.00 | 33.30 | **41.70** |
| hard | semantic | MultiPlate | 15 | 0.00 | 44.40 | **33.30** |
| hard | execution | PositionChangeTo | 11 | 0.00 | 75.00 | **33.30** |
| hard | execution | PositionChangeTo | 13 | 0.00 | 10.00 | **33.30** |
| medium | semantic | MainCarrot | 18 | 50.00 | 68.80 | **81.20** |
| medium | semantic | Plate | 14 | 50.00 | 80.00 | **100.00** |
| medium | semantic | MainCarrot | 19 | 48.60 | 68.60 | **77.10** |
| medium | semantic | MainCarrot | 16 | 48.30 | 75.90 | **89.70** |
| medium | execution | Position | 5 | 47.80 | 73.90 | **87.00** |
| medium | visual | VisionTexture05 | 7 | 47.10 | 76.50 | **76.50** |
| medium | visual | VisionWhole05 | 3 | 47.10 | 76.50 | **100.00** |
| medium | execution | EEPose | 10 | 46.70 | 60.00 | **73.30** |
| medium | visual | VisionImage | 14 | 46.20 | 92.30 | **100.00** |

| Difficulty | Category | Variant | Task | SFT | PPO | Ours |
|---|---|---|---|---|---|---|
| medium | visual | VisionImage | 11 | 44.40 | 66.70 | **100.00** |
| medium | semantic | Plate | 12 | 43.80 | 68.80 | **100.00** |
| medium | execution | Position | 9 | 43.80 | 25.00 | **62.50** |
| medium | execution | EEPose | 7 | 43.80 | 37.50 | **50.00** |
| medium | execution | EEPose | 9 | 42.90 | 42.90 | **71.40** |
| medium | visual | VisionWhole05 | 5 | 42.90 | 78.60 | **78.60** |
| medium | visual | VisionWhole05 | 9 | 42.10 | 57.90 | **73.70** |
| medium | visual | VisionTexture05 | 15 | 41.20 | 52.90 | **58.80** |
| medium | visual | VisionWhole03 | 15 | 41.20 | 52.90 | **64.70** |
| medium | execution | Position | 8 | 40.00 | 66.70 | **73.30** |
| medium | visual | VisionImage | 12 | 40.00 | 85.00 | **95.00** |
| medium | visual | VisionTexture05 | 14 | 40.00 | 80.00 | **100.00** |
| medium | visual | VisionWhole03 | 14 | 40.00 | 90.00 | **90.00** |
| medium | execution | PositionChangeTo | 0 | 38.10 | 57.10 | **71.40** |
| medium | semantic | MainCarrot | 20 | 37.50 | 95.80 | **100.00** |
| medium | visual | VisionWhole05 | 12 | 37.50 | 68.80 | **81.20** |
| medium | execution | EEPose | 14 | 36.80 | 73.70 | **84.20** |
| medium | visual | VisionWhole03 | 8 | 35.30 | 88.20 | **88.20** |
| medium | execution | Position | 14 | 35.00 | 65.00 | **75.00** |
| medium | semantic | Plate | 11 | 33.30 | 66.70 | **73.30** |
| medium | execution | Position | 3 | 33.30 | 53.30 | **46.70** |
| medium | execution | EEPose | 6 | 33.30 | 77.80 | **77.80** |
| medium | visual | VisionTexture03 | 11 | 33.30 | 73.30 | **100.00** |
| medium | visual | VisionWhole05 | 10 | 33.30 | 46.70 | **40.00** |
| medium | execution | EEPose | 15 | 31.60 | 21.10 | **31.60** |
| medium | semantic | Plate | 6 | 30.80 | 69.20 | **84.60** |
| medium | visual | VisionTexture03 | 6 | 30.80 | 100.00 | **100.00** |
| medium | visual | VisionWhole05 | 6 | 30.80 | 53.80 | **100.00** |
| medium | semantic | MainCarrot | 23 | 30.40 | 34.80 | **56.50** |
| medium | execution | EEPose | 4 | 29.40 | 76.50 | **88.20** |
| medium | execution | EEPose | 12 | 29.40 | 88.20 | **100.00** |
| medium | visual | VisionWhole05 | 15 | 29.40 | 29.40 | **47.10** |
| medium | semantic | MultiCarrot | 19 | 29.20 | 33.30 | **37.50** |
| medium | semantic | MultiCarrot | 16 | 28.60 | 35.70 | **54.50** |
| medium | execution | EEPose | 8 | 28.60 | 66.70 | **71.40** |
| medium | visual | VisionTexture03 | 5 | 28.60 | 78.60 | **100.00** |
| medium | visual | VisionTexture05 | 5 | 28.60 | 57.10 | **75.00** |
| medium | visual | VisionWhole03 | 2 | 27.80 | 90.50 | **81.00** |
| medium | execution | Position | 15 | 27.30 | 27.30 | **36.40** |
| medium | visual | VisionImage | 1 | 26.70 | 33.30 | **66.70** |
| medium | semantic | MainCarrot | 24 | 26.10 | 65.20 | **73.90** |
| medium | semantic | MultiCarrot | 17 | 25.00 | 50.00 | **57.10** |
| medium | execution | PositionChangeTo | 5 | 25.00 | 75.00 | **87.50** |
| medium | visual | VisionTexture05 | 12 | 25.00 | 75.00 | **81.20** |
| medium | semantic | MultiCarrot | 20 | 24.20 | 66.70 | **75.80** |
| medium | execution | Position | 6 | 23.50 | 47.10 | **88.20** |
| medium | execution | Position | 13 | 23.50 | 35.30 | **58.80** |
| medium | visual | VisionTexture05 | 3 | 23.50 | 82.40 | **100.00** |
| medium | execution | PositionChangeTo | 15 | 23.10 | 34.60 | **30.80** |
| medium | visual | VisionTexture05 | 6 | 23.10 | 69.20 | **100.00** |
| easy | visual | VisionImage | 13 | 92.30 | 61.50 | **76.90** |

| Difficulty | Category | Variant | Task | SFT | PPO | Ours |
|---|---|---|---|---|---|---|
| easy | visual | VisionTexture03 | 0 | 88.90 | 100.00 | **100.00** |
| easy | visual | VisionWhole03 | 0 | 88.20 | 88.90 | **100.00** |
| easy | visual | VisionImage | 7 | 87.00 | 82.60 | **95.70** |
| easy | visual | VisionImage | 8 | 86.70 | 86.70 | **100.00** |
| easy | visual | VisionImage | 0 | 84.60 | 84.60 | **96.20** |
| easy | semantic | Plate | 0 | 83.30 | 88.90 | **94.40** |
| easy | visual | VisionTexture05 | 0 | 83.30 | 83.30 | **75.00** |
| easy | visual | VisionWhole03 | 10 | 76.90 | 66.70 | **80.00** |
| easy | visual | VisionTexture03 | 3 | 76.50 | 94.10 | **100.00** |
| easy | visual | VisionImage | 10 | 75.00 | 58.30 | **66.70** |
| easy | visual | VisionWhole03 | 9 | 75.00 | 78.90 | **73.70** |
| easy | semantic | Plate | 10 | 73.30 | 33.30 | **46.70** |
| easy | execution | Position | 4 | 73.30 | 20.00 | **73.30** |
| easy | visual | VisionImage | 15 | 73.30 | 46.70 | **80.00** |
| easy | visual | VisionImage | 5 | 71.40 | 100.00 | **100.00** |
| easy | visual | VisionImage | 4 | 70.60 | 94.10 | **100.00** |
| easy | visual | VisionTexture03 | 7 | 70.60 | 76.50 | **100.00** |
| easy | visual | VisionImage | 3 | 70.00 | 90.00 | **100.00** |
| easy | visual | VisionTexture03 | 14 | 70.00 | 90.00 | **100.00** |
| easy | semantic | Plate | 9 | 68.40 | 57.90 | **68.40** |
| easy | visual | VisionTexture03 | 10 | 66.70 | 40.00 | **80.00** |
| easy | visual | VisionWhole05 | 0 | 66.70 | 72.20 | **75.00** |
| easy | semantic | Plate | 3 | 64.70 | 94.10 | **94.10** |
| easy | semantic | Plate | 7 | 64.70 | 70.60 | **88.20** |
| easy | visual | VisionTexture03 | 8 | 64.70 | 100.00 | **100.00** |
| easy | visual | VisionTexture05 | 8 | 64.70 | 94.10 | **100.00** |
| easy | visual | VisionWhole05 | 8 | 64.70 | 70.60 | **83.30** |
| easy | execution | Position | 0 | 64.30 | 50.00 | **57.10** |
| easy | visual | VisionImage | 9 | 64.30 | 50.00 | **100.00** |
| easy | visual | VisionTexture03 | 9 | 63.20 | 47.40 | **80.00** |
| easy | visual | VisionTexture05 | 9 | 63.20 | 68.40 | **73.70** |
| easy | visual | VisionWhole03 | 7 | 62.50 | 82.40 | **100.00** |
| easy | visual | VisionWhole03 | 12 | 62.50 | 93.80 | **93.80** |
| easy | semantic | Plate | 13 | 60.00 | 50.00 | **80.00** |
| easy | visual | VisionTexture05 | 13 | 60.00 | 40.00 | **80.00** |
| easy | visual | VisionWhole03 | 13 | 60.00 | 70.00 | **60.00** |
| easy | semantic | Plate | 8 | 58.80 | 94.10 | **94.10** |
| easy | execution | Position | 7 | 58.30 | 41.70 | **75.00** |
| easy | execution | EEPose | 13 | 58.30 | 41.70 | **66.70** |
| easy | semantic | Plate | 5 | 57.10 | 100.00 | **100.00** |
| easy | visual | VisionImage | 6 | 57.10 | 85.70 | **92.90** |
| easy | visual | VisionWhole03 | 5 | 57.10 | 100.00 | **100.00** |
| easy | semantic | Plate | 4 | 56.20 | 93.80 | **93.80** |
| easy | visual | VisionTexture03 | 4 | 56.20 | 100.00 | **100.00** |
| easy | visual | VisionWhole05 | 4 | 56.20 | 81.20 | **100.00** |
| easy | execution | EEPose | 0 | 53.80 | 84.60 | **92.30** |
| easy | visual | VisionTexture05 | 10 | 53.30 | 26.70 | **53.30** |
| easy | visual | VisionWhole03 | 4 | 53.30 | 93.80 | **100.00** |
| easy | semantic | Plate | 15 | 52.90 | 41.20 | **58.80** |
| easy | visual | VisionImage | 2 | 52.90 | 88.20 | **88.20** |
| easy | visual | VisionTexture03 | 15 | 52.90 | 52.90 | **41.20** |
| easy | visual | VisionWhole03 | 3 | 52.90 | 94.10 | **100.00** |

*(Table XI continued)*

| Difficulty | Category | Variant | Task | SFT | PPO | Ours |
|---|---|---|---|---|---|---|
| easy | visual | VisionWhole05 | 7 | 52.90 | 70.60 | **88.20** |
| easy | semantic | MainCarrot | 17 | 52.00 | 64.00 | **92.00** |
| easy | execution | Position | 10 | 50.00 | 25.00 | **50.00** |
| easy | execution | EEPose | 2 | 50.00 | 64.30 | **64.30** |
| easy | visual | VisionTexture03 | 13 | 50.00 | 50.00 | **90.00** |
| easy | visual | VisionTexture05 | 4 | 50.00 | 93.80 | **100.00** |
| easy | visual | VisionWhole05 | 13 | 50.00 | 40.00 | **60.00** |

This appendix provides implementation details for the language-model experiments in Section V-H. We use the same PDE interface as in the VLA experiments: each problem $g$ has a canonical prompt $p_g$, PDE proposes an alternative prompt $p \sim \rho(\cdot \mid g, \mathcal{H})$ for training, and evaluation is always performed under $p_g$.

### H. Experiment: AIME 2026

**Setup.** We evaluate on AIME 2026, which contains 30 olympiad-style math problems, each with a unique answer given as a three-digit integer between 000 and 999. We fine-tune Qwen3-4B with GRPO, using Claude Sonnet 4.6 as the prompt sampler $\rho$. Training runs for $T = 3$ PDE iterations. At each iteration, we sample $K = 1$ alternative prompt per problem, generate $N = 8$ rollouts per prompt, and run 10 epochs of GRPO. Both GRPO and GRPO+PDE use learning rate $3 \cdot 10^{-6}$.

**Implementation details.** For the first PDE iteration, we query the prompt sampler $\rho$ once per problem $g$ to obtain an alternative prompt $p$. In later iterations, we reuse $p$ if it outperforms the canonical prompt $p_g$ after the previous training iteration. Otherwise, we query $\rho$ for a new prompt, conditioning on the canonical prompt, the previous alternative prompt, and the rollout history $\mathcal{H}$. Thus, prompt admission is determined adaptively by comparison to the canonical prompt rather than by a fixed threshold $\eta$.

For mixed backpropagation, this experiment uses an arithmetic-mixture variant:

$$\log \pi_{\mathrm{mix}}(a \mid o) = \log(\pi_\theta(a \mid o, p_g) + \pi_\theta(a \mid o, p)),$$

up to an additive constant. This differs from the geometric-mean form used in the VLA experiments.

**Evaluation and results.** At each checkpoint, we evaluate under the canonical prompt $p_g$ only. For each problem, we generate 20 independent rollouts and report accuracy over all $30 \times 20 = 600$ generations. Figure 7 compares GRPO and GRPO+PDE. GRPO+PDE improves early learning, reaching 48.9% accuracy after the first iteration compared with 45.8% for GRPO, and both methods reach 53.3% after three iterations.

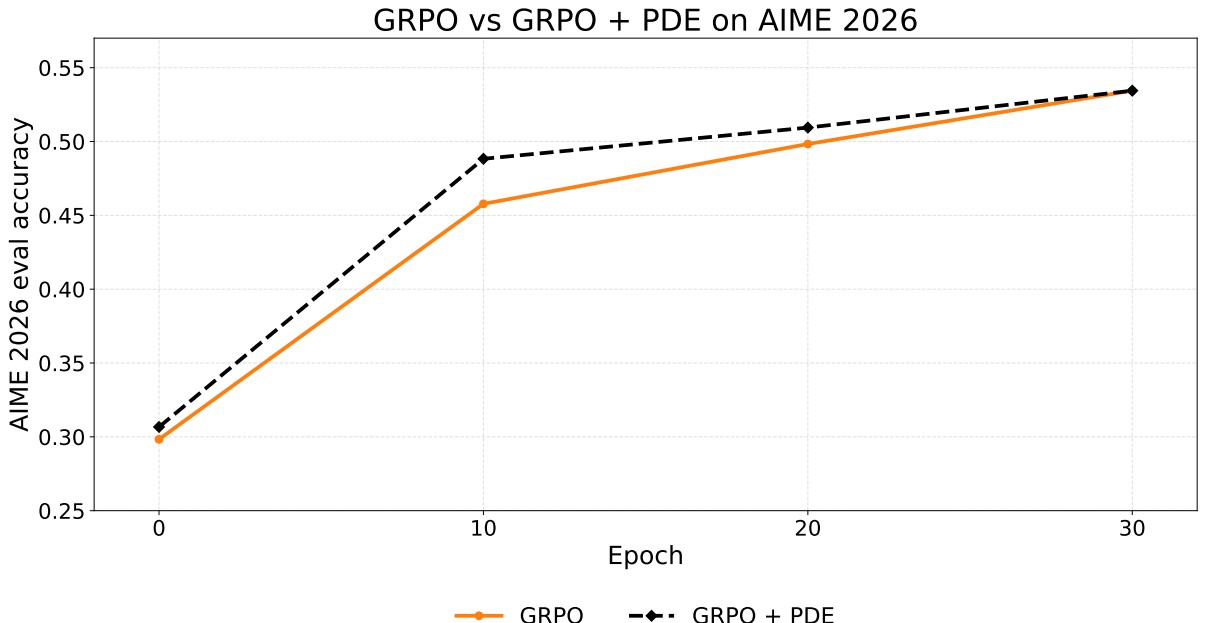

Fig. 7: AIME 2026 accuracy for GRPO and GRPO+PDE. Evaluation uses only the canonical problem prompt $p_g$.

## I. Experiment: LiveCodeBench

**Setup.** We evaluate on LiveCodeBench, which contains 1,055 competitive-programming problems with hidden tests. We use 600 problems for training and reserve the remaining 455 for held-out evaluation. We fine-tune with RLOO at learning rate $5 \cdot 10^{-5}$.

**Implementation details.** For each eligible training problem $g$, the policy itself serves as the prompt sampler $\rho$. The policy first generates 10 candidate solutions under the canonical prompt $p_g$. It is then re-prompted to produce an alternative prompt $p$ conditioned on $p_g$ and its own raw rollouts. The sampler observes the rollouts but not their rewards. This refinement loop is run for 3 rounds, and the best alternative prompt is cached. During training, the cached prompt is used alongside the canonical prompt in the RLOO update. The first 50% of training problems are marked as eligible for prompt refinement; held-out evaluation uses only the canonical prompt $p_g$.

**Evaluation and results.** At each batch checkpoint, we evaluate on the first 100 held-out problems by generating one greedy rollout per problem and reporting the fraction of solutions that pass all hidden tests. The final checkpoint is evaluated in the same way on 400 held-out problems. Figure 8 compares RLOO and PDE+RLOO. PDE+RLOO learns substantially faster: it reaches 50% held-out accuracy by batch 4, while RLOO reaches this level around batch 7. PDE+RLOO also improves early accuracy, achieving 42.4% vs. 29.9% at batch 1, 53.1% vs. 36.0% at batch 3, and 54.6% vs. 41.3% at batch 4. The two methods converge later in training, showing that PDE primarily improves early sample efficiency by exposing useful solution modes sooner.

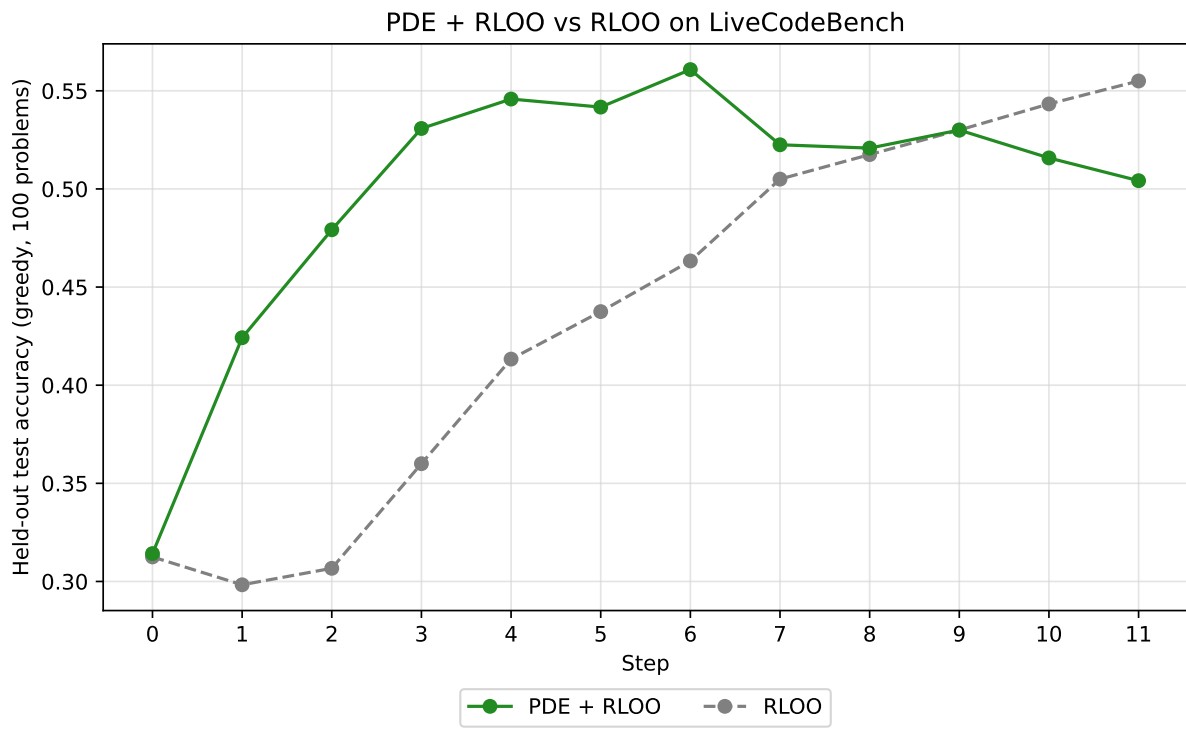

Fig. 8: LiveCodeBench held-out accuracy for RLOO and PDE+RLOO. Training uses alternative prompts from PDE, while evaluation uses only the canonical prompt $p_g$.