# OpenReview forum: "Prompt-Driven Exploration"
_roboticsfoundation.org/RSS/2026/Workshop/RL4VLA — RL4VLA_

### Official Review · Reviewer_3nG9 · 2026-06-23

**Rating:** 7
**Confidence:** 3

**Review:**

# Strengths

#### 1. Core insight is novel and well explained
   Using prompt-space as an exploration axis is a novel idea, and the benefit of engineering prompts to condition entire rollouts in order to make perturbations global is well-explained and illustrated via examples.

#### 2. Zero-reward bootstrapping result
   Strong illustrative result, with success rate going from 0% to 98% while action-noise remains close to zero.

#### 3. Mixed backpropogation
  Modified PPO ratio is a smart solution, seems to give genuinely good results in the ablation table.

#### 4. Breadth of evaluation
The evaluation covers various robot benchmark tests (LIBERO-PRO, ManiSkill, real-world manipulation) as well as language model-based reinforcement learning tasks, illustrating that proposed exploration strategy is not confined to any one particular setting or architecture.

# Weaknesses

#### 1. Breakdown of VLM query cost
  Per-task, about 50 rollouts are run for just the prompt-discovery stage, plus VLM inference on each. A breakdown of "time spent querying VLM" vs "time spent running the robot", illustrating the VLM overhead would provide an important insight. A breakdown of wall-clock training time or VLM-query cost relative to environment interaction would help clarify the practical trade-offs.

#### 2. Comparison to CoVer-VLA
   While a reasonable practical excuse has been provided for exclusion, at minimum a qualitative comparison of the two approaches would help strengthen the related works section.

#### 3. Posterior Sampling
   The posterior-sampling interpretation seems mainly conceptual. The method does not explicitly keep a posterior distribution over prompts or policies, nor does it carry out a Bayesian update. Therefore, the VLM-based prompt generation mechanism is likely better described as an adaptive prompt-search approach inspired by posterior sampling, rather than a direct implementation of it.

---

### Official Review · Reviewer_1y47 · 2026-06-27
**Prompt Drive Exploration: Exploring by changing input prompts to a prompt-conditioned policy in an online RL training setup**

**Rating:** 7
**Confidence:** 4

**Review:**

The paper proposes using prompts to seed the behavior of a prompt-conditioned policy, driving global behavior changes necessary for exploration and autonomous improvement during online RL. Overall, the paper is very well written with strong results on the simulation suite.

It would be good to understand the gains and limitations of this setup on real world robots as part of future work.

---

### Decision · Program_Chairs · 2026-07-03

**Decision:**

Accept

**Comment:**

This paper studies prompt-conditioned exploration for RL fine-tuning of VLAs, showing strong results in simulation and some real robot tasks. The reviewers' main concerns are related to unclear VLM cost, limited comparison to related work, and how posterior sampling is interpreted. Despite these issues, we view the paper as a valuable workshop contribution. For the camera-ready version, the authors should improve the comparison to CoVer-VLA and clarify the posterior sampling view.